# Naturally occurring variation in a cytochrome P450 modifies thiabendazole responses independently of beta-tubulin

J. B. Collins[1], Clayton M. Dilks[2,3], Steffen R. Hahnel[2¤], Briana Rodriguez[2], Bennett W. Fox[4], Elizabeth Redman[5], Jingfang Yu[4], Brittany Cooke[6,7], Kateryna Sihuta[6,7], Mostafa Zamanian[8], Peter J. Roy[6,7,9], Frank C. Schroeder[4], John S. Gilleard[5], Erik C. Andersen[1] *

1 Department of Biology, Johns Hopkins University, Baltimore, Maryland, United States of America, 2 Molecular Biosciences, Northwestern University, Evanston, Illinois, United States of America, 3 Interdisciplinary Biological Sciences Program, Northwestern University, Evanston, Illinois, United States of America, 4 Boyce Thompson Institute and Department of Chemistry and Chemical Biology, Cornell University, Ithaca, New York, United States of America, 5 Department of Comparative Biology and Experimental Medicine, University of Calgary, Calgary, Canada, 6 Department of Molecular Genetics, University of Toronto, Toronto, Canada, 7 The Donnelly Centre for Cellular and Biomolecular Research, University of Toronto, Toronto, Canada, 8 Department of Pathobiological Sciences, University of Wisconsin, Madison, Wisconsin, United States of America, 9 Department of Pharmacology and Toxicology, University of Toronto, Toronto, Canada

¤ Current address: Boehringer Ingelheim Vetmedica GmbH, Rhein, Germany
* erik.andersen@gmail.com

**Data Availability Statement:** Scripts and data for this study are available at https://github.com/AndersenLab/2024_cyp35D1_TBZ_manuscript.

## Abstract

Widespread anthelmintic resistance has complicated the management of parasitic nematodes. Resistance to the benzimidazole (BZ) drug class is nearly ubiquitous in many species and is associated with mutations in beta-tubulin genes. However, mutations in beta-tubulin alone do not fully explain all BZ resistance. We performed a genome-wide association study using a genetically diverse panel of *Caenorhabditis elegans* strains to identify loci that contribute to resistance to the BZ drug thiabendazole (TBZ). We identified a quantitative trait locus (QTL) on chromosome V independent of all beta-tubulin genes and overlapping with two promising candidate genes, the cytochrome P450 gene *cyp-35D1* and the nuclear hormone receptor *nhr-176*. Both genes were previously demonstrated to play a role in TBZ metabolism. NHR-176 binds TBZ and induces the expression of CYP-35D1, which metabolizes TBZ. We generated single gene deletions of *cyp-35D1* and *nhr-176* and found that both genes play a role in TBZ response. A predicted high-impact lysine-to-glutamate substitution at position 267 (K267E) in CYP-35D1 was identified in a sensitive strain, and reciprocal allele replacement strains in different genetic backgrounds were used to show that the lysine allele conferred increased TBZ resistance. Using competitive fitness assays, we found that neither allele was deleterious, but the lysine allele was selected in the presence of TBZ. Additionally, we found that the lysine allele significantly increased the rate of TBZ metabolism compared to the glutamate allele. Moreover, yeast expression assays showed that the lysine version of CYP-35D1 had twice the enzymatic activity of the glutamate allele. To connect our results to parasitic nematodes, we analyzed four *Haemonchus contortus* cytochrome P450 orthologs but did not find variation at the 267 position in fenbendazole-

**Funding:** This work was supported by the National Institutes of Health NIAID grant R01AI153088 to ECA. The funders had no role in study design, data collection and analysis, decision to publish, or preparation of the manuscript. JBC was supported, in part, by the National Institutes of Health NIAID grant R01AI153088 to ECA.

**Competing interests:** The authors have declared that no competing interests exist.

resistant populations. Overall, we confirmed that variation in this cytochrome P450 gene is the first locus independent of beta-tubulin to play a role in BZ resistance.

## Author summary

Benzimidazoles (BZs) are the most common drug class used to control parasitic nematodes, but because of overuse, resistance is widespread. The known genetic causes of BZ resistance are associated with mutations in beta-tubulin and are the most well understood of any anthelmintic class. However, BZ response varies significantly and differential levels of resistance likely require mutations in genes independent of beta-tubulin. We used the free-living model nematode *Caenorhabditis elegans* to identify and characterize a novel cytochrome P450 gene, *cyp-35D1*, associated with natural resistance to the BZ drug thiabendazole (TBZ). We demonstrated that a lysine at position 267 confers TBZ resistance and was selected over multiple generations of TBZ treatment. This allele significantly increased the rate of TBZ metabolism in both *C. elegans* and yeast. In conclusion, we have characterized the role of variation in a cytochrome P450 that contributes to TBZ resistance, independent of mutations in beta-tubulin.

## Introduction

Parasitic nematode infections pose a significant health risk to humans and livestock around the globe. An estimated 1.3 billion people are infected with at least one soil-transmitted helminth species, causing significant socio-economic burdens and heavily impacting quality of life [1]. In livestock production, infections are often subclinical but cause significant economic losses, reaching as high as 14% of the production value in some species [2]. Benzimidazoles (BZs) are among the most common anthelmintics used in human and veterinary medicine. Overreliance and misuse of BZs have led to widespread resistance in veterinary medicine [3,4]. Although not yet widespread in human helminth infections, studies indicate that resistance to anthelmintics is emerging and represents a significant concern [5–7]. Understanding the mechanisms of anthelmintic resistance represents one of the most critical steps in parasite control.

A cycle of discovery, in which candidate genes and mutations are identified using experiments in the free-living nematode *Caenorhabditis elegans* and then validated in parasites, or vice versa, has informed almost the entire body of work for anthelmintic resistance in nematodes [8]. Although improved genetic resources for the ruminant parasitic nematode *Haemonchus contortus* have emerged in recent years [9,10], this parasite requires a host, which makes high-throughput genetic mappings and genome editing to confirm the phenotypic effects of putative resistance mutations impractical. The genetic diversity in parasite populations can be mimicked using natural populations of *C. elegans*. Using the cycle of discovery, the beta-tubulin genes *ben-1* [11] and *Hco-tbb-iso-1* were validated as the primary targets of BZs in *C. elegans* and *H. contortus*, respectively [12–14]. Since initially identifying the target of BZs, nine resistance alleles have been discovered in parasites and validated in *C. elegans*: Q134H, F167Y, E198A, E198I, E198K, E198L, E198T, E198V, E198Stop, and F200Y [13,15–19]. However, mutations in beta-tubulin alone do not explain all phenotypic variation in BZ response in parasite populations, and research into natural variation in *C. elegans* has

identified multiple genomic regions containing genes that impact BZ responses independently of beta-tubulin genes [4,20–22].

Here, we leveraged a large collection of *C. elegans* wild strains [23,24] and recombinant lines [25] to investigate natural variation in response to thiabendazole (TBZ), a BZ anthelmintic [26]. A genome-wide association study (GWAS) and linkage mapping (LM) experiment identified a genomic region on chromosome V that was significantly correlated with differential responses to TBZ. We further narrowed this region to two candidate genes: the cytochrome P450 *cyp-35D1* and the nuclear hormone receptor *nhr-176*. Both of which have previously been shown to play a role in TBZ metabolism [27,28]. Interestingly, nuclear hormone receptors are often associated with neurons and sensory responses [29]. However, NHR-176 is associated with the nematode intestine where TBZ binds to the receptor and then induces the expression of *cyp-35D1* [27,28]. The CYP-35D1 enzyme then initiates the hydroxylation-dependent metabolism of TBZ. We used CRISPR-Cas9 genome editing to generate deletions of *cyp-35D1* and *nhr-176*, which conferred significant susceptibility, confirming the role of both genes in the TBZ response. No *nhr-176* variants were identified in the parental strains used for linkage mapping, but we identified a lysine-to-glutamate substitution at position 267 (K267E) in CYP-35D1, indicating that *cyp-35D1* likely underlies differences in TBZ response. Using competitive fitness assays, we found that the lysine allele did not affect fitness in control conditions but was significantly favored in TBZ conditions, reaching near fixation after seven generations. Then, we measured the abundances of three key metabolites in the metabolism of TBZ: hydroxylated TBZ (TBZ-OH), TBZ-*O*-glucoside, and TBZ-*O*-phosphoglucoside and found significant differences in the accumulation of metabolites. Yeast expressing the CYP-35D1 lysine allele and exposed to TBZ were found to be almost twice as efficient at metabolizing TBZ compared to the enzyme with the glutamate allele, confirming that the lysine allele confers resistance by increased TBZ metabolism. Using deep amplicon sequencing analysis of similar variant sites in orthologous CYP genes of fenbendazole-resistant *H. contortus* populations, we found no amino acid variation, suggesting that the BZ response is not affected by variation at position 267 of *H. contortus* CYP-35D1 orthologs. We also investigated the potential evolutionary history of alleles at position 267 of CYP-35D1 in wild *C. elegans* populations to determine if the variation is correlated with the global distribution of wild strains. We found that the BZ-resistant lysine allele is represented by a single, globally distributed haplotype, which is likely a recent gain of greater resistance to TBZ. Overall, we characterized how natural variation in CYP-35D1 contributes to resistance, representing the first gene independent of beta-tubulin to be associated with BZ resistance.

## Results

### Two genes on chromosome V contribute to natural variation in *C. elegans* TBZ response independent of beta-tubulin genes

To quantify variation in TBZ response, we measured phenotypes of 214 wild *C. elegans* strains after exposure to 62.5 μM TBZ in a previously developed high-throughput assay based on nematode body length as a proxy for development [20,25,30,31]. Because TBZ inhibits development, shorter body lengths indicate slower development and therefore represent greater susceptibility to TBZ [16,20], similar to traditional parasitic nematode larval development assays. Highly TBZ-resistant strains had predicted loss-of-function variation in *ben-1* (Fig 1A), as previously shown [20].

Variation in *ben-1* is known to play a large role in BZ response, but our goal was to identify novel resistance mechanisms independent of the beta-tubulin genes (*tbb-1*, *tbb-2*, *mec-7*, *tbb-4*, *ben-1*, *tbb-6*). We performed genome-wide association (GWAS) mappings using the TBZ

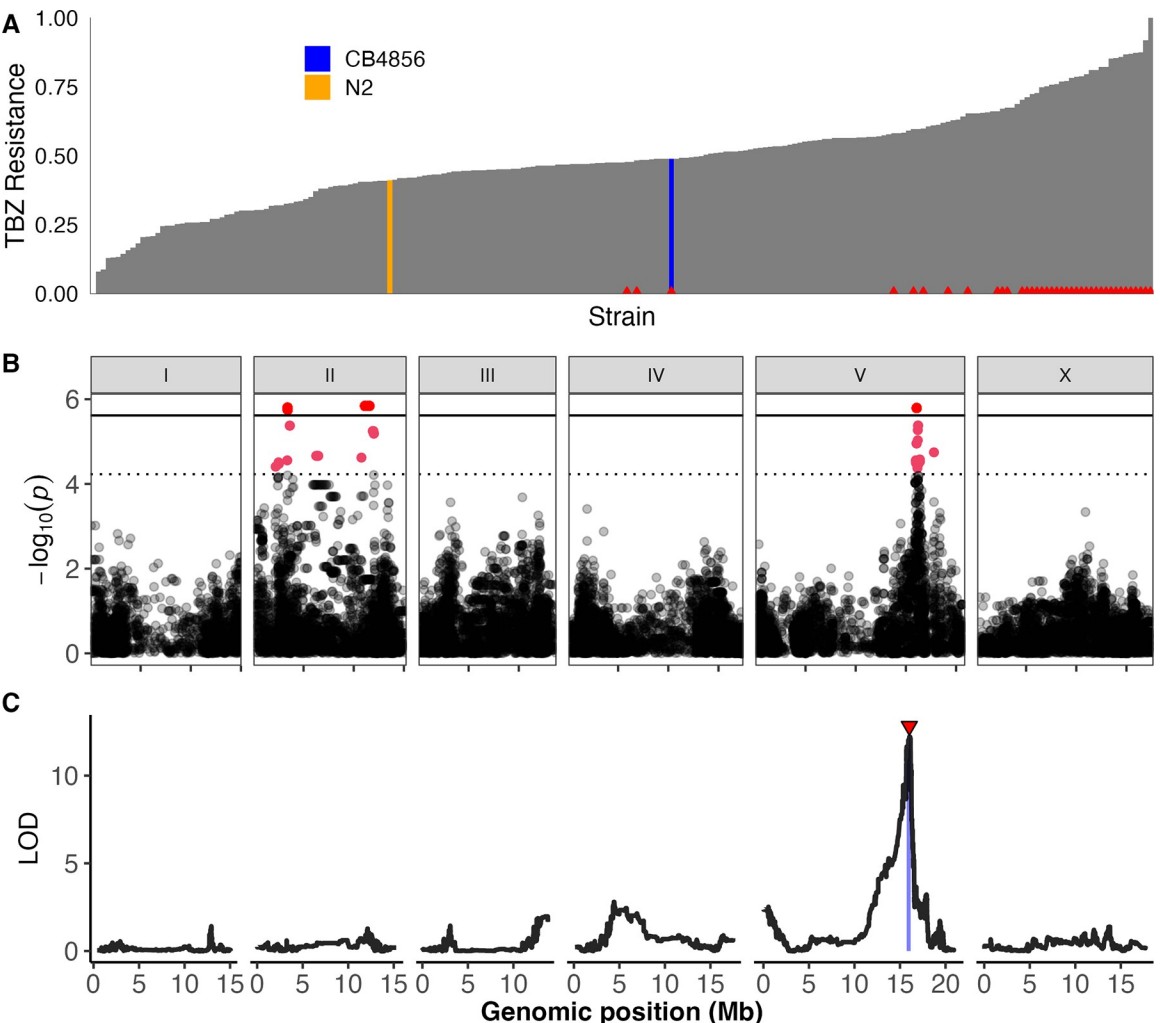

**Fig 1. One large-effect QTL on chromosome V underlies differences in TBZ response.** (A) Distribution of normalized relative resistance in 62.5 μM TBZ is shown in order from most susceptible to most resistant for 214 wild *C. elegans* strains. Strains with variation in *ben-1* have a red triangle at the base of the bar for that strain. (B) Genome-wide association mapping results for animal length that has been regressed for the effect of *ben-1* are shown. The genomic position is shown on the x-axis, and statistical significance (-log10(*p*) values) is shown on the y-axis for each SNV. SNVs are colored pink if they pass the Eigen significance threshold (dashed horizontal line) or red if they pass the Bonferroni significance threshold (solid horizontal line). (C) Linkage mapping results for animal length are shown. The genomic position is shown on the x-axis, and the statistical significance (logarithm of the odds (LOD) score) is shown on the y-axis for 13,003 genomic markers. A red triangle indicates a significant QTL, and a blue rectangle indicates the 95% confidence interval around the QTL.

response data and two quantitative trait loci (QTL) were identified, one on the left of chromosome II and another on the right of chromosome V (Figs 1B and S1A) [20]. Neither QTL overlapped with the six known beta-tubulin genes. Next, we regressed the effects of *ben-1* variation on the TBZ response data to identify novel genes that could have been hidden because of the strong effects of *ben-1* and found only the QTL on the right of chromosome V (S1B Fig). The lack of the chromosome II QTL in the *ben-1*-regressed mapping suggests that this QTL is associated with variation in *ben-1*, which has also been previously documented in mappings using albendazole [20], so we focused on the chromosome V QTL. Analysis of the peak marker on chromosome V showed that strains matching the reference genotype were significantly more resistant than strains with an alternative genotype (S2A Fig).

In addition to the GWAS, we performed linkage mapping with 219 recombinant inbred advanced intercross lines (RIAILs) generated by a cross of the laboratory strain N2 and the wild strain CB4856 [25] at 32.5 μM TBZ and found a single significant QTL on the right of chromosome V that overlaps with the QTL identified in the GWAS (Fig 1C). We looked at the difference between recombinant lines with the reference or alternative alleles at the peak marker and found that lines with the N2 reference allele were significantly more resistant than lines with the CB4856 alternate allele (S2B Fig). We then backcrossed RIAILs with the parental strains (*i.e.*, N2 or CB4856) to create near-isogenic lines (NILs) that had the chromosome V region of one genetic background introgressed into the opposite genetic background to confirm that the interval on chromosome V was responsible for the difference in phenotype. The NIL strain ECA238 has the N2 genetic background at all loci except the QTL on chromosome V, and the NIL strain ECA239 has the CB4856 genetic background at all loci except the QTL on chromosome V (S3A and S3B Fig). When exposed to 32.5 μM TBZ, the ECA238 strain was significantly more susceptible than the N2 strain, whereas the ECA239 strain was significantly more resistant than the CB4856 strain (S3C Fig), validating that the QTL on chromosome V underlies differential responses to TBZ. Fine mapping of the QTL on chromosome V identified two genes highly correlated with resistance to TBZ, *cyp-35D1* and *nhr-176* (S4 Fig). Previous studies found that both *cyp-35D1* and *nhr-176* play a role in TBZ response, where TBZ binds to NHR-176 and induces expression of *cyp-35D1*, which encodes a cytochrome P450 that metabolizes TBZ [27,28]. Although the genes are adjacent in the genome, they are not part of an operon and were examined separately. Because both genes play a role in the metabolism of TBZ, we measured the effects of the deletion of *cyp-35D1* and *nhr-176*, both individually and together, in both the N2 and CB4856 genetic backgrounds to further investigate the role of each gene in the TBZ response phenotype. Deletions of each gene were exposed to 32.5 μM TBZ. The deletion of *cyp-35D1* conferred susceptibility in both genetic backgrounds (S5 Fig). Deletion of the nuclear hormone receptor conferred an equivalent level of susceptibility compared to the deletion of *cyp-35D1* (S5 Fig). These results show that both genetic backgrounds have functional genes for *cyp-35D1* and *nhr-176*. Furthermore, deletion of both genes together did not significantly alter responses compared to the deletion of *nhr-176* alone (S6 Fig), in agreement with the known requirement for *nhr-176* in the expression of *cyp-35D1*.

## Natural variation in *cyp-35D1* underlies differences in *C. elegans* responses to TBZ

Multiple variants with minor allele frequencies greater than 5% were found in both *cyp-35D1* and *nhr-176* in strains from the GWAS population. However, linkage mapping identified an overlapping QTL in strains where only a lysine-to-glutamate substitution at position 267 in CYP-35D1 was found. Because linkage mapping pinpointed a correlation only with *cyp-35D1*, we hypothesized that this cytochrome P450 underlies the observed differences in TBZ response. To test the effect of each allele at position 267 of CYP-35D1, allele-replacement strains between the N2 and CB4856 strains were generated using CRISPR-Cas9 genome editing. Two strains in the N2 genetic background were made with glutamate at position 267, and two strains in the CB4856 background were made with lysine at position 267. Responses to 32.5 μM TBZ were tested in the allele-replacement and parental strains (Figs 2 and S7). The strain with the N2 genetic background and the glutamate allele (PHX2701) was significantly more susceptible to TBZ than the N2 parental strain with the lysine allele (Fig 2). The strain with the CB4856 genetic background and the lysine allele (PHX2883) was significantly more resistant to TBZ than the CB4856 parental strain with the glutamate allele, confirming that the lysine at position 267 was sufficient to confer greater levels of TBZ resistance.

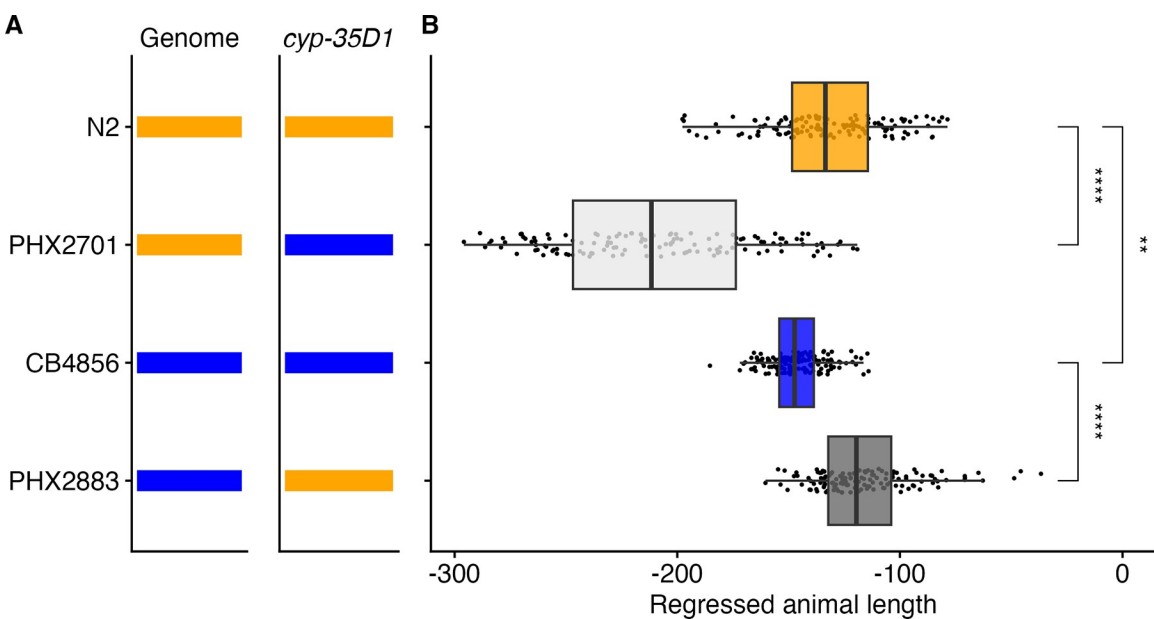

**Fig 2. A lysine at position 267 of CYP-35D1 confers increased resistance to TBZ.** (A) Strain names are displayed on the y-axis. The genomic background and the background where each *cyp-35D1* allele originates are shown as orange or blue for N2 or CB4856, respectively. (B) Regressed median animal length values of response to 32.5 μM TBZ are shown on the x-axis. Each point represents a well that contains approximately 30 animals after 48 hours of exposure to TBZ. Data are shown as box plots with the median as a solid vertical line, the right and left vertical lines of the box represent the 75th and 25th quartiles, respectively. The top and bottom horizontal whiskers extend to the maximum point within 1.5 interquartile range from the 75th and 25th quartiles, respectively. Statistical significance is shown above each strain comparison; the N2 and CB4856 strain values are also significantly different ($p < 0.01$ = **, $p < 0.0001$ = ****, Tukey HSD).

In addition to the lysine-to-glutamate substitution identified in the CB4856 strain, a second lysine-to-aspartate substitution at position 267 (K267D) was observed in wild strains that were found to be more susceptible than strains with the glutamate allele (S8 Fig). We generated allele replacement strains between the N2 strain and the DL238 strain, which harbors the K267D allele, and compared TBZ response between the parental lines, as well as to the glutamate and aspartate replacement strains in the N2 background. Aspartate as position 267 did not alter the TBZ response in the N2 background. However, the DL238 strain was found to be highly susceptible compared to all of the other strains (S9 Fig), indicating that susceptibility in DL238 is not likely mediated solely by the allele at position 267 of CYP-35D1.

## The lysine allele is not deleterious in the absence of TBZ

To determine if the resistant lysine allele causes any negative effects on organismal fitness in the absence of TBZ, we conducted competitive fitness assays in the N2 genetic background. Changes in allele frequency over seven generations (Fig 3A and 3C) were used to calculate the relative fitness of strains with a lysine or glutamate allele at position 267 in CYP-35D1. Relative fitness did not differ in control conditions, indicating that neither the lysine nor glutamate allele conferred any deleterious consequences in the absence of TBZ selective pressure (Fig 3B). However, the lysine allele was found to be significantly more fit than the glutamate allele in the presence of TBZ (Fig 3D). The benefits in the presence of TBZ and the lack of deleterious effects in the absence of TBZ indicate that, once present in a population, the lysine allele would be selected after TBZ exposure and likely maintained in the absence of selection pressure.

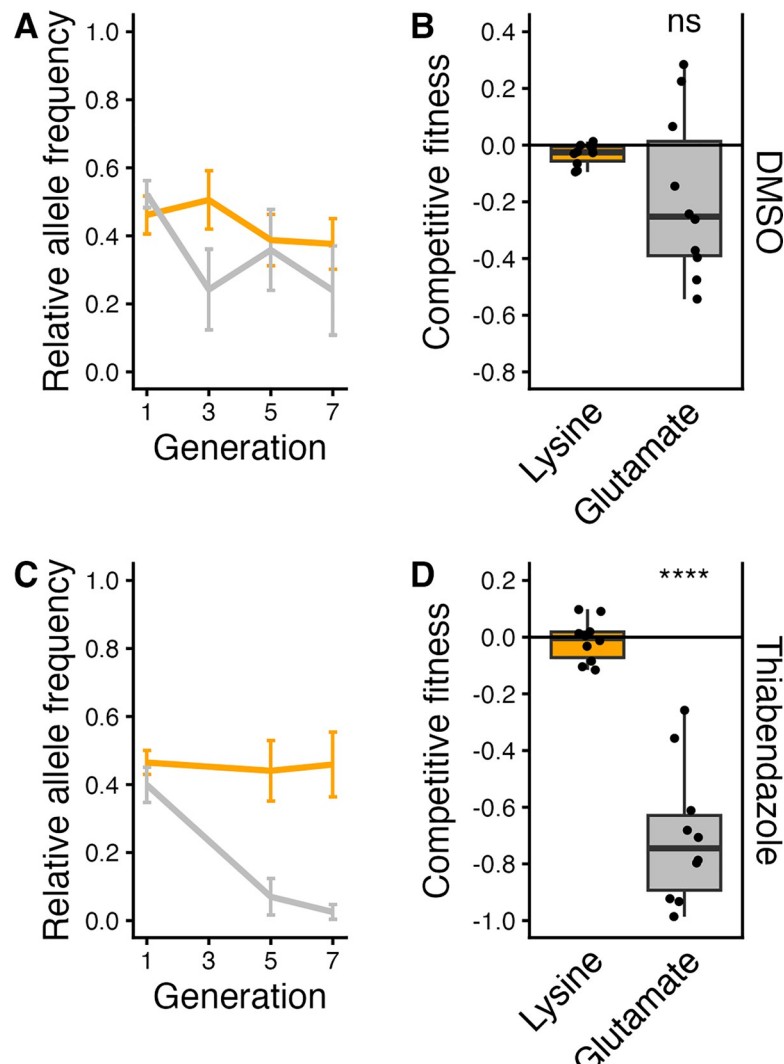

**Fig 3. Competitive fitness assay across seven generations in DMSO and TBZ.** (A) The change in allele frequencies of the lysine (orange) and glutamate (gray) alleles in the N2 background was determined using competitions between a barcoded N2 strain in 1% DMSO. Generation is shown on the x-axis, and the relative allele frequency of each strain is shown on the y-axis. (B) The log2-transformed competitive fitness of each allele is plotted. The allele tested is shown on the x-axis, and the competitive fitness is shown on the y-axis. Each point represents a biological replicate of that competition experiment. (C) The change in allele frequencies of the lysine (orange) and glutamate (gray) alleles in the N2 background was determined using a competition with a barcoded N2 strain in 25 μM TBZ. (D) The log2-transformed competitive fitness value of each allele is plotted. Each point represents one biological replicate of the competition assay. Data are shown as box plots, with the median as a solid horizontal line and the top and bottom of the box representing the 75th and 25th quartiles, respectively. The top and bottom whiskers are extended to the maximum point that is within 1.5 interquartile range from the 75th and 25th quartiles, respectively. The top and bottom vertical whiskers extend to the maximum point within 1.5 interquartile range from the 75th and 25th quartiles, respectively. Significant differences between the wild-type strain and all other alleles are shown as asterisks above the data from each strain ($p > 0.05$ = ns, $p < 0.0001$ = ****, Tukey HSD).

## A natural variant in CYP-35D1 affects the metabolism of TBZ

Metabolism of TBZ is initiated by cytochrome P450-dependent oxidation of the benzimidazole ring producing TBZ-hydroxide (TBZ-OH), which can be glycosylated to TBZ-O-glucose (TBZ-O-Glu) and phosphorylated to TBZ-O-phosphoglucoside (TBZ-O-PGlu) to aid in

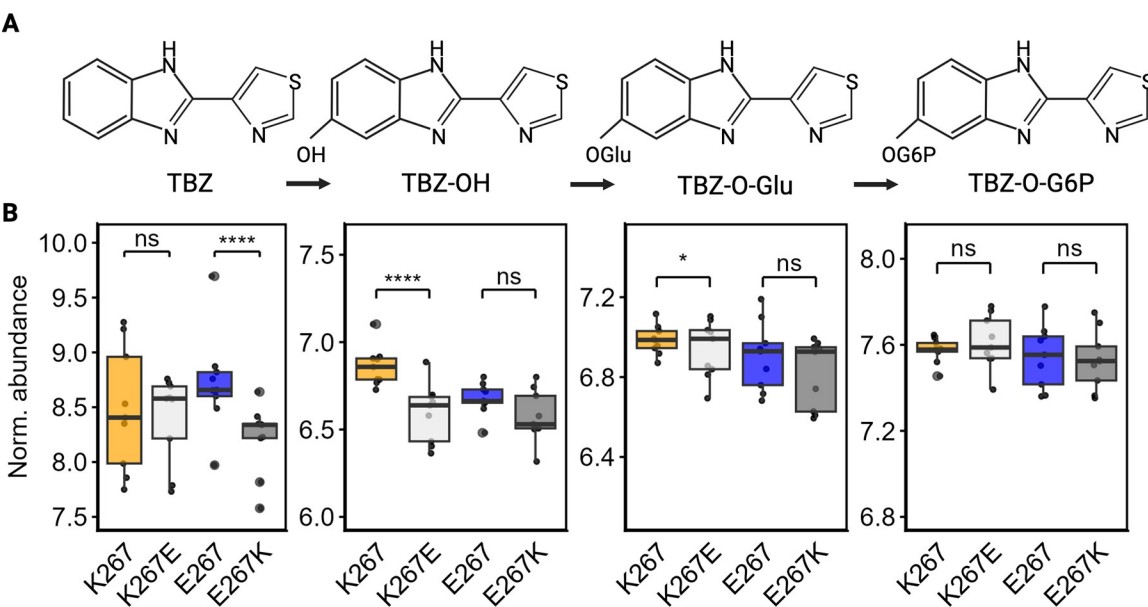

**Fig 4. The abundances of TBZ and TBZ metabolites in the endo-metabolome six hours after exposure.** (A) Simplified TBZ metabolic pathway. (B) The change in the normalized abundances of TBZ and three metabolites: TBZ-OH, TBZ-O-glucoside, and TBZ-O-phosphoglucoside) are shown, with samples taken at six hours after exposure to 50 μM TBZ. CYP-35D1 alleles are shown on the x-axis with K267 (orange) and K267E (light gray) in the N2 genetic background, and E267 (blue) and E267K (dark gray) in the CB4856 genetic background. Normalized metabolite abundance is shown on the y-axis. Abundances are shown as the log of abundance after normalization to the abundance of ascr#2. Each point represents an individual replicate. Data are shown as box plots, with the median as a solid horizontal line and the top and bottom of the box representing the 75th and 25th quartiles, respectively. The top and bottom whiskers are extended to the maximum point that is within 1.5 interquartile range from the 75th and 25th quartiles, respectively. The top and bottom vertical whiskers extend to the maximum point within 1.5 interquartile range from the 75th and 25th quartiles, respectively. Statistical significance between strains with the same genetic background at the same time point is shown ($p > 0.05$ = ns, $p < 0.05$ = *, $p < 0.0001$ = ****, Wilcoxon Rank Sum test with Bonferroni correction).

elimination by efflux enzymes [27]. To investigate the metabolic effects of CYP-35D1 variation on the metabolism of TBZ, we measured the abundances of TBZ metabolites inside the animals (endo-metabolome) (Figs 4 and S10) and in the conditioned medium (exo-metabolome) (S11 Fig) at two and six hours of TBZ exposure using high performance liquid chromatography coupled to high-resolution mass spectrometry (HPLC-HRMS). At both two and six hours of exposure, the abundance of TBZ in the endo-metabolome of the E267K allele in the CB4856 background was reduced relative to the parental strain, suggesting increased metabolism to downstream metabolites in the E267K edited strain (Figs 4 and S10A). On the other hand, K267E in the N2 background did not impact the amount of TBZ retained in the endo-metabolome as compared to its parental strain. No significant differences in metabolite abundance were found at either time point in the exo-metabolome among the strains, suggesting the involvement of additional detoxification and excretion mechanisms not affected by this single missense mutation.

To determine if the observed metabolic effects could be recapitulated in another model system, we generated strains of *Saccharomyces cerevisiae* (yeast), a system previously used to measure metabolism of cyprocides [32], that express *C. elegans* lysine or glutamate versions of CYP-35D1. These yeast strains were exposed to 100 μM TBZ for six hours before analysis of TBZ-OH abundance using liquid chromatography-quadrupole time-of-flight (LC-QTOF). We found that the lysine version of CYP-35D1 was approximately 1.8 times more efficient at metabolizing TBZ to TBZ-OH than the glutamate version (Fig 5). We found that when the

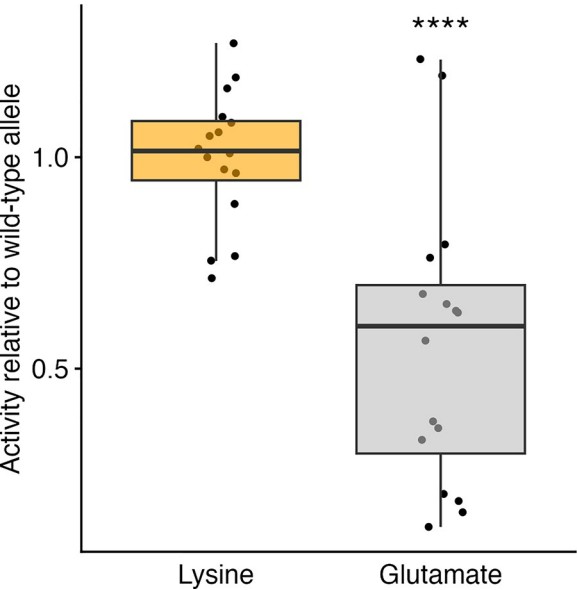

**Fig 5. Activity of wild-type and mutant CYP-35D1 expressed in yeast.** The metabolic activity of the lysine and glutamate versions of CYP-35D1 when expressed in yeast and exposed to 100 μM TBZ for six hours is shown. The activity of each enzyme is shown relative to the wild-type activity. Each point represents an individual replicate. Data are shown as box plots, with the median as a solid horizontal line and the top and bottom of the box representing the 75th and 25th quartiles, respectively. The top and bottom whiskers are extended to the maximum point that is within 1.5 interquartile range from the 75th and 25th quartiles, respectively. The top and bottom vertical whiskers extend to the maximum point within 1.5 interquartile range from the 75th and 25th quartiles, respectively. Significant differences between the lysine and glutamate allele are shown as asterisks ($p > 0.05$ = ns, $p < 0.05$ = *, Wilcoxon Rank Sum test with Bonferroni correction). Differences observed between replicates of the same transgenic yeast line exceed any observed differences between different lines.

lysine and glutamate alleles are compared within the same genetic background, the lysine allele significantly increased the metabolism and excretion of TBZ and TBZ metabolites.

## Evolutionary history and global distribution of CYP-35D1 alleles

We next investigated the evolutionary history and global distribution of the three alleles at position 267 (K267, K267E, and K267D) in CYP-35D1 to determine if the variants are regionally distributed, suggesting an isolated selection event, or globally distributed, suggesting multiple selection events for TBZ resistance. Using data available from the *Caenorhabditis* Natural Diversity Resource [23], we found that all three alleles are broadly distributed and sampled from the same geographical areas as well as across multiple continents (Fig 6A) [23]. We investigated the haplotypes at the *cyp-35D1* locus and generated a tree for a region containing 25 kb to either side of *cyp-35D1* (Fig 6B). The tree could be divided into three distinct clades, with the lysine allele in two of the clades. Tajima's D was calculated for the same region used to generate the tree, and found no signature of selection at the locus (S12 Fig).

Additionally, we investigated *cyp-35D1* orthologs in other *Caenorhabditis* species (Fig 7A). For two members of the *elegans* super-group, *C. briggsae* and *C. brenneri*, the genes *Cbr-cyp-35D1* and *CAEBREN_29747*, respectively, are the most phylogenetically similar orthologs of *C. elegans cyp-35D1*. For *C. tropicalis*, another member of the *elegans* group, the closest ortholog *Csp11.Scaffold629.g12881* is more closely related to orthologs from the *japonica* group species *C. becei* and *C. panamensis*. Position 267 of CYP-35D1 and related orthologs do not have the TBZ-resistant lysine allele in any orthologous gene across the *Caenorhabditis* genus (Fig 7B).

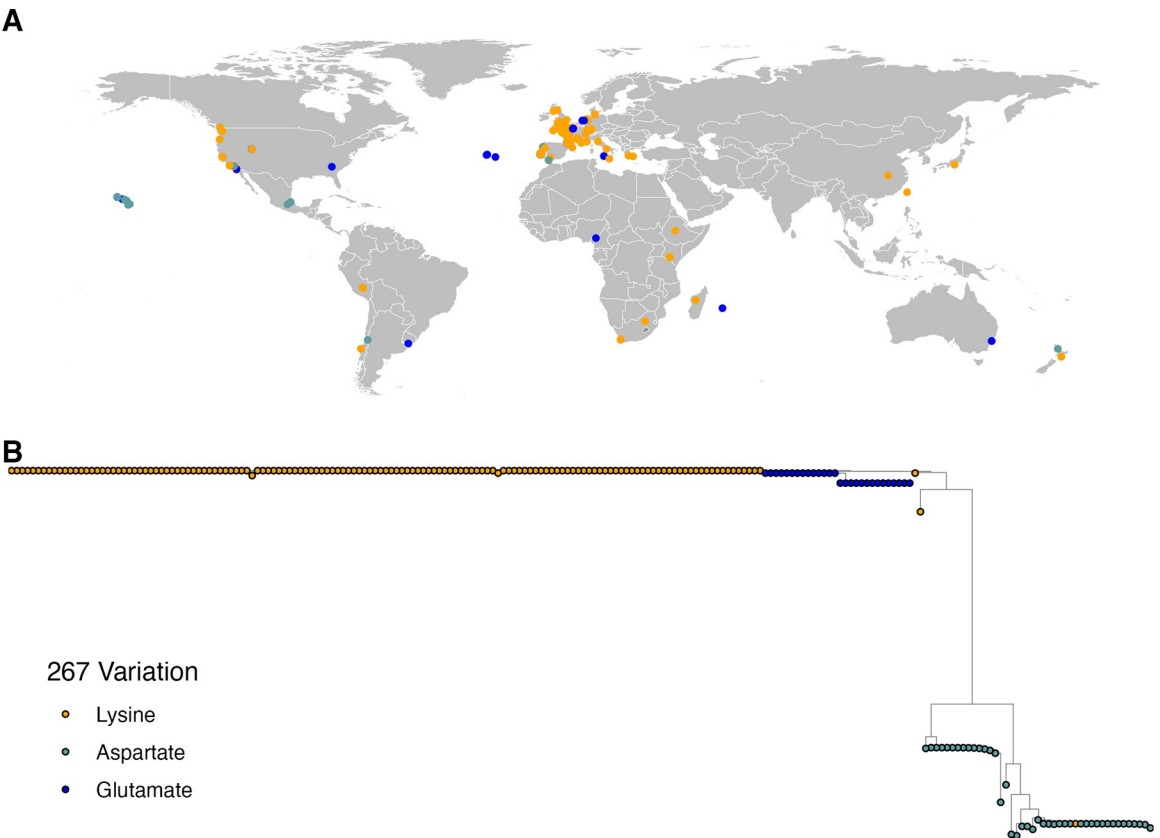

**Fig 6. Variation in CYP-35D1 is not regionally distributed.** A map with the locations where each wild strain was recovered is shown. Each point represents a strain isolation location and is colored by the allele at the 267 position of CYP-35D1. (B) A neighbor-joining tree for the cyp-35D1 locus is shown. Each circle represents one wild strain and is colored by the allele at the 267 amino-acid position. Some areas where many strains were collected could not show all points because they are covered by other points. Map generated using ggplot2 under a MIT license (https://ggplot2.tidyverse.org/LICENSE.html).

Glutamate and aspartate are the most common alleles identified in the *Caenorhabditis* species, except for serine in *C. bovis* and asparagine in *C. becei*. The four most closely related *H. contortus* orthologs also did not have a lysine at position 267 but did have aspartate or glutamate (Fig 7B) [33].

To determine if resistant populations of *H. contortus* contained natural variation in *cyp-35D1* orthologs, we performed deep-amplicon sequencing of the four most closely related orthologs from 128 archived samples of fenbendazole-resistant *H. contortus* and focused on the 267 position. Similar to what was observed in *Caenorhabditis* species, an acidic residue was present in 100% of the populations from three of the orthologs, and asparagine was found to be ubiquitous in the fourth ortholog (Table 1). The lack of variation at position 267 in FBZ-resistant *H. contortus* indicates a lack of selection within these populations and might represent differences in the metabolism of FBZ compared to TBZ.

## Discussion

Over the last 30 years, beta-tubulin mutations have been associated with BZ resistance in both free-living and parasitic nematode species [11–13,16,17,20]. However, mutations in beta-tubulin genes alone do not explain all of the observed degrees of BZ resistance and additional loci have been identified in *C. elegans* [20,21]. Understanding other mechanisms of resistance can

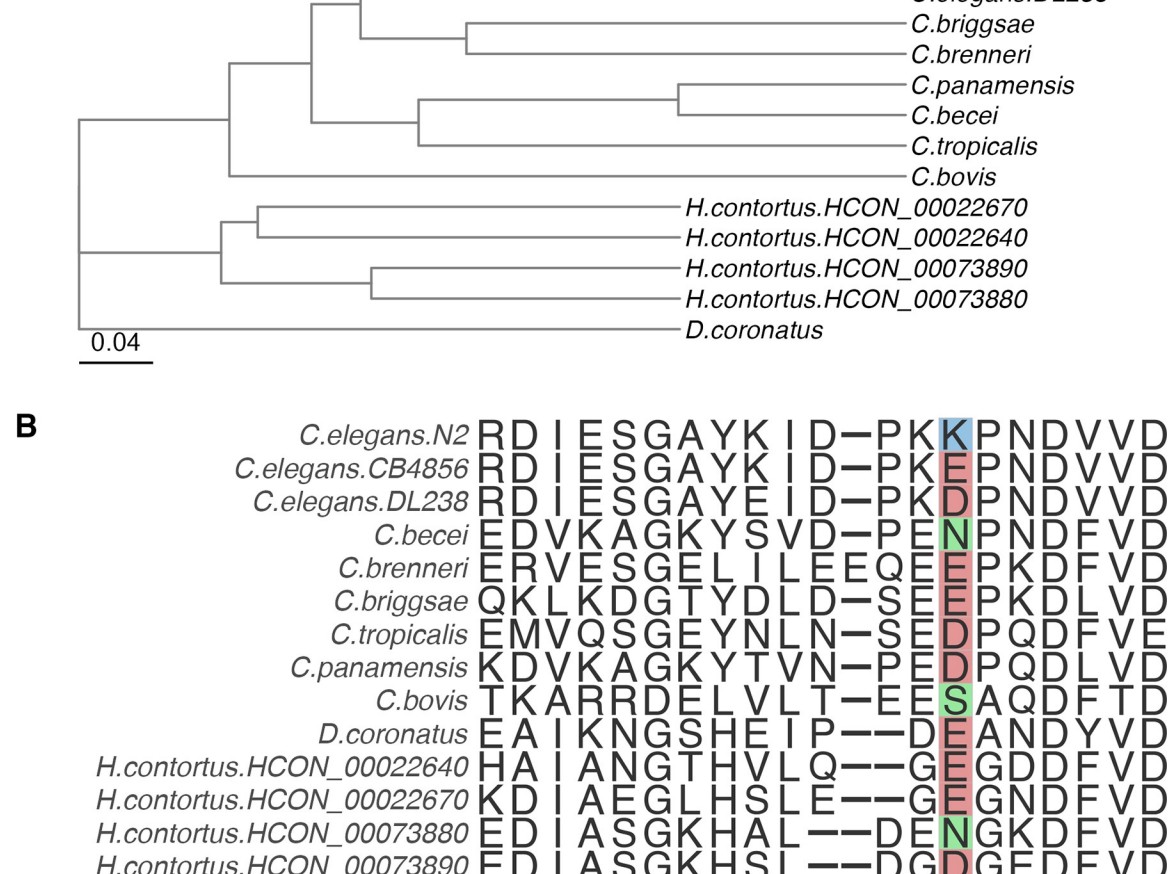

**Fig 7. Lysine at position 267 of CYP-35D1 orthologs is unique to *C. elegans*.** Neighbor-joining tree for seven species across the *Caenorhabditis* clade, four orthologs from *H. contortus*, and the free-living nematode *Diploscapter coronatus* as an outgroup. The tree scale is denoted on the left side of the tree and represents differences in the sequences of the CYP-35D1 ortholog found in each species. (B) Amino-acid alignment of the region surrounding position 267 in CYP-35D1 for the species shown in the neighbor-joining tree.

provide potential targets for novel treatments, improve efficacy by reducing the level of selection, slow the development of resistance, and lead to the development of more comprehensive diagnostics for BZ resistance. Many gene families associated with xenobiotic response have been suggested as potential mediators of BZ resistance, including UDP-glycosyltransferases, P-glycoproteins, and cytochrome P450s [34,35]. Here, we have leveraged the natural diversity

**Table 1. No variation exists at position 267 in the four *H. contortus* orthologs of CYP-35D1.** Deep amplicon sequencing of 128 populations of fenbendazole-resistant *H. contortus* was performed to search for variation at position 267 of the four most closely related orthologs of *C. elegans* CYP-35D1. The number of samples that were successfully sequenced and analyzed varied between orthologs, with the number for each ortholog shown in parenthesis.

| | Allele at Position 267 (% of Samples (n)) | | |
|---|---|---|---|
| | **Glutamate** | **Asparagine** | **Aspartate** |
| *Hc_00022640* | 100% (110) | 0% (0) | 0% (0) |
| *Hc_00022670* | 100% (110) | 0% (0) | 0% (0) |
| *Hc_00073880* | 0% (0) | 100% (118) | 0% (0) |
| *Hc_00073890* | 0% (0) | 0% (0) | 100% (103) |

and robust genomic toolkit of *C. elegans* to identify and characterize natural variation in a cytochrome P450, CYP-35D1, that modifies *C. elegans* responses to TBZ independently of beta-tubulin.

Selection for resistance alleles in populations comes from exposure to xenobiotic compounds. Certain prokaryotes have been found to produce natural BZ derivatives as part of a vitamin B12 synthesis pathway [36], and contamination with commercial BZs used in agriculture has been found to be common and associated with lengthy half-lives of compounds like TBZ in the environment, providing a source of BZ selection for nematode populations [37,38]. Interestingly, the evolution of the more active CYP-35D1 enzyme appears to predate commercial use of BZs, because the lysine allele was identified in the N2 strain, which was first isolated in 1951, 18 years prior to the release of TBZ. Natural BZs in the environment have likely led to selection for the resistant allele in CYP-35D1, corresponding with the significantly higher relative fitness of the lysine allele compared to the glutamate allele in treatment conditions. Similar to previous studies of the effects of *ben-1* mutations on competitive fitness [16], we found that the fitness of strains with the resistance allele are not significantly selected against in control conditions, suggesting that once the allele emerges in a population, it will be maintained in the absence of selection. We do not know if *cyp-35D1* resistance is dominant, and future experiments can address this point. Resistance phenotypes conferred by *ben-1* were previously shown to be recessive [17].

In addition to measuring how the lysine allele affects competitive fitness, we also wanted to determine how a single amino acid change alters TBZ metabolism. As previously noted, genes associated with xenobiotic response have long been considered as mediators of anthelmintic resistance, where animals that are more efficient at the breakdown and elimination of the drug have a competitive advantage. Metabolic analysis of the lysine and glutamate versions of CYP-35D1 in both *C. elegans* and yeast found that the efficiency of TBZ metabolism is significantly increased by the lysine allele. More rapid metabolism agrees with the observed resistance conferred by the lysine allele, because faster metabolism would cause reduced exposure to the more active form of the drug, decreasing treatment efficacy. The confirmation of natural variation in a metabolic gene that alters drug response highlights the importance of studying gene families that play a role in xenobiotic response [39]. Insights gained from studying xenobiotic response can lead to the development of treatments that target and alter metabolic pathways, providing a novel means of improving treatment efficacy and slowing the spread of resistance.

Lastly, we examined the evolutionary history of the lysine allele in *Caenorhabditis* species, as well as if the allele is found in parasite populations. The broader geographical distribution of the glutamate allele, along with its high frequency across nematode orthologs of CYP-35D1, suggests that glutamate represents the ancestral allele, and the lysine allele is likely a more recent adaptation. The significant number of strains carrying lysine at position 267 indicates that this allele is selected and, once selected, will likely persist in populations. Beyond *C. elegans* alone, amino acid sequences from the reference sequence of each non-*elegans* species and that of three *C. elegans* strains (N2, CB4856, DL238) used in the current study were analyzed for the allele at position 267. Only *C. elegans* contained the basic lysine allele at position 267, whereas the other species had acidic or neutral residues, likely representing the ancestral state. However, it is important to note that other nematode species like *Strongyloides stercoralis* and *Bursaphelenchus okinawaensis* also have lysine at position 267 in predicted *cyp-35D1* orthologs. Additional variants at this position are also found in predicted *cyp-35D1* orthologs in other species, but the high levels of conservation of cytochrome P450s make functional determination of differences difficult. Therefore, we focused on *C. elegans* variants where we could mechanistically test the effects of each allele and search in the available FBZ-resistant *H. contortus* populations. Future efforts can explore the role of the cytochrome P450 family in BZ responses.

The presence of lysine only in *C. elegans* indicates an evolutionarily more recent acquisition of this resistance allele, but it is important to note that we only examined a small region surrounding position 267 of CYP-35D1. Data available in the *Caenorhabditis* Natural Diversity Resource show multiple CYP-35D1 variants that alter amino acid residues in both *C. briggsae* and *C. tropicalis* [23]. Although we can draw no conclusions about the resistance status, alleles at other positions can alter responses in a similar manner to the lysine allele at position 267. Therefore, the lack of variation at position 267 does not preclude the possibility that additional variants that alter BZ responses are present in other species. Also, although we did not identify variation in the nuclear hormone receptor NHR-176 in the CB4856 strain, further exploration of the second gene in the QTL would be interesting. Variants in *nhr-176* found in other strains are present and could alter the TBZ response. Additionally, differences in expression of *cyp-35D1* and *nhr-176* are not found between CB4856 and N2. However, expression differences in other strains could also play a role in altering the TBZ response.

When orthologs of CYP-35D1 were examined in the parasite *H. contortus*, no basic residues were identified within the four orthologs, similar to the non-*elegans Caenorhabditis* species. However, the reference genome represents a single population, and variation between populations must be examined to make broader conclusions. Analysis of position 267 in 128 *H. contortus* populations that were exposed to long-term FBZ selection in the field was performed to determine if any variation in that residue exists. No variation in position 267 was identified in any of the four orthologs. However, it is important to note that TBZ has not been used in the field for many decades, caused in part by the creation of more effective BZ compounds. The tested *H. contortus* populations have been exposed to heavy selection with FBZ. Furthermore, previous GWAS mappings using FBZ and ABZ did not identify the *cyp-35D1* QTL, although other novel QTL were identified in these mappings (Zamanian et al. 2018; Hahnel et al. 2018) [21,20]. Structural differences could alter how each drug is metabolized. For example, TBZ contains a thiazole ring that increases the susceptibility of TBZ to oxidative metabolism. Metabolism of TBZ and FBZ could use different pathways, so variation associated with TBZ resistance might not be present in FBZ-resistant samples. In addition, as noted among *Caenorhabditis* species, we only examined a single amino acid site in four *H. contortus* cytochrome P450s, and alteration of TBZ responses could be caused by variation at other sites or in other cytochrome P450s. To fully determine if similar variation can occur in parasite populations, identification of populations that have undergone TBZ selection pressure is needed to determine if variation is present at CYP-35D1 orthologs of parasites.

## Conclusions

Despite the lack of variation found in *H. contortus* orthologs, the role of variation in CYP-35D1 in TBZ response reinforces the need to study how variation in drug metabolism genes modifies anthelmintic responses. Although the effects of variation in CYP-35D1 are relatively small compared to the impact of variation in the beta-tubulin target of BZs, changes to drug metabolism could have significant impacts on the evolution of resistance in a susceptible population. Inadequate dosing is all too common, and the increased metabolism conferred by variation in a metabolic pathway could enable parasite survival until a larger effect mutation is acquired, such as mutations in beta-tubulin. Conversely, in populations where a mutation in beta-tubulin has reached fixation, a change in a metabolic pathway could further increase resistance levels. In either situation, variation in a gene involved in a BZ metabolic pathway could have a significant impact on the development and severity of resistance. By studying and leveraging knowledge of drug metabolism, treatments can be designed that inhibit metabolism and increase treatment efficacy, promoting the sustainability of current treatments and slowing the development of resistance.

## Methods

### *C. elegans* strains

Nematodes were grown on plates of modified nematode growth media (NGMA) containing 1% agar and 0.7% agarose and seeded with OP50 bacteria [40]. Plates were maintained at 20˚C for the duration of all experiments. Before each assay, animals were grown for three generations to reduce the multigenerational effects of starvation. A total of 214 wild strains were phenotyped for genome-wide association mapping. For linkage mapping, 219 recombinant inbred advanced intercross lines (RIAILs) were generated from a cross between an N2 strain, with the CB4856 *npr-1* allele and a transposon insertion in the *peel-1* gene (QX1430), and the Hawaiian wild strain CB4856 [25]. Near-isogenic lines (NILs) were generated by backcrossing a RIAIL of interest to a parent strain for several generations using PCR amplicons flanking insertion-deletion (indels) variants to track the introgressed region [30]. The NILs were whole-genome sequenced after backcrossing to verify that only the desired introgressed region was present. All strains (S1 File) are available and maintained by the *Caenorhabditis* Natural Diversity Resource (CaeNDR) [23].

CRISPR-Cas9-edited strains were generated within the lab or by Suny Biotech (Fuzhou, China). Suny Biotech generated the allele-replacement strains with the N2 (PD1074) background (PHX2701, PHX2702) and the CB4856 background (PHX2882, and PHX2883). Additional strains with single gene deletions of either *cyp-35D1* or *nhr-176*, deletions of both genes, as well as strains with a K267D substitution, were generated using CRISPR-Cas9 genome editing as previously described [16,20]. All oligonucleotides used for CRISPR-Cas9 editing and confirmation are found in S18 File. After injection, possibly edited strains underwent two generations of confirmation using Sanger sequencing, ensuring that strains were homozygous for the desired genotype. Two independent edits were generated to control for any potential off-target effects of editing.

### Genome-wide association mapping

We phenotyped 214 wild strains of *C. elegans* in both DMSO and TBZ conditions as described previously [20]. Briefly, strains were passaged for three generations post-starvation on NGMA plates to alleviate multi-generational starvation effects. After passage, populations of each strain were bleach synchronized (Protocol available at https://andersenlab.org/Research/Protocols/) in triplicate to control for variation caused by bleach effects. Approximately 50 embryos were resuspended in 50 μL of K medium [41] and dispensed into 96-well plates and allowed to arrest overnight. The following day, arrested L1 larvae (first larval stage) were fed lyophilized bacterial lysate (*E. coli* HB101 strain) at 5 mg/mL in K medium. Nematodes were grown for 48 hours at 20˚C with constant shaking at 180 rpm. Three L4 larvae (fourth larval stage) were sorted into a 96-well plate with 10 mg/mL of bacterial lysate, 50 μM kanamycin, and either 1% DMSO or 62.5 μM TBZ dissolved in 1% DMSO using a large-particle flow cytometer (COPAS BIOSORT, Union Biometrica; Holliston, MA). Animals were grown for 96 hours at 20˚C with constant shaking at 180 rpm. The animals and their offspring were treated with sodium azide (50 mM in M9 buffer) to straighten the animals for accurate length measurements with the COPAS BIOSORT. Measurements collected in the high-throughput fitness assay were processed with the *easysorter* R (4.0.3) package [42]. Analysis was performed on a strain-specific basis as previously described [16,30,31].

We performed a genome-wide association mapping using the differences in responses of strains exposed to TBZ and DMSO conditions. The mean for time of flight (time it took for animal to pass through a laser) (mean.TOF) was used as a measure of animal length, and data

were analyzed using the mapping pipeline *NemaScan* (https://github.com/AndersenLab/NemaScan) [43]. The phenotype data were mapped with and without the effects of *ben-1* in the data to identify genes independent of known resistance mechanisms as described previously [20]. Briefly, strains with *ben-1* loss of function mutations were identified and phenotypes were corrected using the following linear model: lm(animallength~(ben−1LoF)). Genotype data for the tested strains were acquired from the 20220216 CaeNDR release. *NemaScan* was run using default parameters in the mappings profile to perform association and fine mappings. Significance thresholds were determined using Bonferroni and Eigen significance values. The Bonferroni threshold is based on the number of markers in the analysis, and the Eigen threshold corrects for the number of genetically independent markers in the data set. The Eigen threshold takes advantage of the extensive linkage disequilibrium in *C. elegans* to limit the number of "unique" markers [44].

## Linkage mapping

219 RIAILs [25] were phenotyped in both DMSO and TBZ (32.5 μM) conditions, as previously done for the genome-wide association study. Linkage mapping was performed on animal length (q90.TOF), as measured with the COPAS BIOSORT and processed with the *easysorter* R (4.0.3) package [42], with the *linkagemapping* (https://github.com/AndersenLab/linkagemapping) *R* package [30,45]. A cross object derived from the whole-genome sequencing data of the RIAILs containing 13,003 single nucleotide variants (SNVs) was merged with RIAIL phenotypes with the *merge_pheno* function with the argument *set* = 2. We used the *fsearch* function, adapted from the *R/qtl* package [46], to calculate the logarithm of the odds (LOD) score for every genetic marker and the animal length (mean.TOF) as $-n(ln(1-R^2)/2ln(10))$ where R is the Pearson correlation coefficient between the genotype of the RIAIL marker and animal length [47]. We calculated a 5% genome-wide error rate by permuting the RIAIL phenotype data 1000 times. We categorized the peak QTL marker as the marker with the highest LOD score over the significance threshold. This marker was then used in the model as a cofactor and the mapping analysis was repeated until no further QTL were detected. We then used the *annotate_lods* function to calculate the effect size of the QTL and determine the 95% confidence intervals as defined by 1.5 LOD drop from the peak marker with the argument *cutoff* = "proximal."

## High-throughput assays of edited strains

A previously described high-throughput fitness assay was used for all TBZ response phenotyping assays [48,49]. In the GWAS, strains were prepared as above. For each assay, each bleach of each strain had 96 replicates. Plates were then sealed with gas permeable sealing film (Fisher Cat #14-222-043), placed in humidity chambers, and incubated overnight at 20°C while shaking at 170 rpm (INFORS HT Multitron shaker). The following morning, arrested L1s were fed using frozen aliquots of HB101 *E. coli* suspended in K medium at an optical density 600 nm (OD$_{600}$) of 100. HB101 aliquots were thawed at room temperature, combined, and diluted to OD$_{600}$30 with K medium, and kanamycin was added at a concentration of 150 μM to inhibit further bacterial growth and prevent contamination. Final well concentration of HB101, prepared as above, with kanamycin was OD10 and 50 μM, respectively, and each well was treated with either 1% DMSO or 32.5 μM TBZ in 1% DMSO. Animals were grown for 48 hours with constant shaking, after which, animals were treated with 50 mM sodium azide in M9 buffer to straighten the animals. Following 10 minutes of exposure to sodium azide, each plate was imaged using a Molecular Devices ImageXpress Nano microscope (Molecular Devices, San Jose, CA) with a 2X objective. Images were then processed using CellProfiler (https://github.

com/AndersenLab/CellProfiler) and analyzed using *easyXpress* [48] to obtain animal lengths. Data were normalized and regressed as done previously [49,50]. Briefly, variation attributable to assay and replicate effects was regressed out using a linear model, and residual values were normalized with respect to the average control phenotype by subtracting the mean phenotype in control conditions from the corresponding phenotype in the TBZ condition. Normalized phenotype measurements were used in all downstream statistical analyses.

## Competition assays

We used a previously established pairwise competition assay to assess organismal fitness [51]. Fitness is measured for seven generations by comparing the allele frequency of a test strain against the allele frequency of a wild-type control. Both strains harbor molecular barcodes to distinguish between the two strains using oligonucleotide probes complementary to each barcode allele. Ten L4 individuals of a test strain were placed onto a single 6 cm NGMA plate along with ten L4 individuals of the PTM229 strain (an N2 strain that contains a synonymous change in the *dpy-10* locus that does not have any growth effects compared to the wild-type laboratory N2 strain) [51]. Ten independent NGMA plates of each competition were prepared for each strain in each test condition, 1% DMSO or 25 μM TBZ in 1% DMSO. Animals were grown for one week and transferred to a new plate of the same condition on a 0.5 cm$^3$ NGMA piece from the starved plate. For generations 1, 3, 5, and 7, the remaining individuals on the starved plate were washed into a 15 mL conical tube with M9 buffer and allowed to settle. The pellet was transferred to labeled 1.7 mL microcentrifuge tubes and stored at -80˚C. DNA was extracted using the DNeasy Blood & Tissue kit (QiagenCatalog #: 69506). We quantified the relative allele frequency of each strain as previously described [51]. In short, a digital droplet PCR (ddPCR) approach with TaqMan probes (Applied Biosciences) was used. Extracted genomic DNA was purified with a Zymo DNA cleanup kit (D4064) and diluted to 1 ng/μL. Using TaqMan probes as described previously [51], the ddPCR was performed with *Eco*RI digestion during thermocycling and quantified with a BioRad QX200 device with standard probe absolute quantification settings. The TaqMan probes selectively bind to the wild-type and edited *dpy-10* alleles, serving as markers to quantify the relative abundance of each experimental strain (wild-type *dpy-10*) and the reference strain (PTM229). Relative allele frequencies of each tested allele were calculated using the QuantaSoft software and default settings. Calculations of relative fitness were calculated by linear regression analysis to fit the data to a one-locus generic selection model [51].

## Sample preparation for HPLC-HRMS

A 6 cm NGMA plate with a starved population was chunked to ten 10 cm NGMA plates for the N2, PHX2702, CB4856, and PHX2883 strains. Following 72 hours of growth, populations were bleach synchronized and diluted to approximately 1 embryo/μL in K medium, and 100 mL of the embryo solution, for each strain, was placed into 500 mL Erlenmeyer flasks, and allowed to hatch overnight with constant shaking at 180 rpm at 20˚C. The following day, the hatched L1s were fed HB101 bacteria at a final concentration of $OD_{600}15$ and were grown for 72 hours [52]. Strains were bleach synchronized again, and 750,000 embryos/strain were placed into 4 L flasks, at a concentration of approximately 1 embryo/μL, and allowed to hatch overnight and then fed as above. After 72 hours of growth, each flask was divided into three replicate flasks containing approximately 250,000 young adult animals. Aliquots of approximately 50,000 animals were removed from each flask, for a total of three replicates per strain, before treatment with TBZ at a final concentration of 50 μM; control cultures were treated with an equivalent volume of DMSO (vehicle). In addition to the initial samples, three

replicate samples were taken for each strain after two or six hours of exposure to TBZ. Aliquots were subject to centrifugation at 254 g for 30 seconds, and then the supernatant was transferred to a new 50 mL conical. Worm pellets were rinsed twice with M9, followed by a single rinse with K medium to remove remaining bacteria. Worm pellets were transferred to 1.7 mL Eppendorf tubes. The supernatant and worm pellet samples were flash-frozen in liquid nitrogen, and then stored at -80˚C prior to extraction.

Worm pellets were lyophilized using a Labconco FreeZone 4.5 system for approximately eight hours, prior to disruption. Dried worm pellets were disrupted in a Spex 1600 MiniG tissue grinder after the addition of two stainless steel grinding balls to each sample. Eppendorf tubes were placed in a Cryoblock (Model 1660) cooled in liquid nitrogen, and samples were disrupted at 1,100 rpm for two cycles of 90 seconds, with cooling in between cycles. 1 ml of methanol was added to each Eppendorf tube, and then samples were briefly vortexed and rocked overnight at room temperature. Eppendorf tubes were centrifuged at 20,000 RCF for five minutes in an Eppendorf 5417R centrifuge. Approximately 900 μL of the resulting supernatant was transferred to a clean 4 mL glass vial, and 800 μL fresh methanol added to the sample. The sample was briefly vortexed, centrifuged as described, and the resulting supernatant was combined in the 4 mL glass vial. The extracts were concentrated to dryness in an SCP250EXP Speedvac Concentrator coupled to an RVT5105 Refrigerated Vapor Trap (Thermo Scientific). The resulting powder was resuspended in 150 μL of methanol, followed by vortex and brief sonication. This solution was subject to centrifugation at 20,000 RCF for 10 minutes to remove the precipitate. The resulting supernatant was transferred to an HPLC vial and analyzed by HPLC-HRMS.

## HPLC-HRMS analysis

Reversed-phase chromatography was performed using a Dionex Ultimate 3000 HPLC system controlled by Chromeleon Software (Thermo Fisher Scientific) and coupled to an Orbitrap Q-Exactive mass spectrometer controlled by Xcalibur software (Thermo Fisher Scientific) equipped with a heated electrospray ionization (HESI-II) probe. Extracts prepared as described above were separated on an Agilent Zorbax Eclipse XDB-C18 column (150 mm x 2.1 mm, particle size 1.8 μm) maintained at 40˚C with a flow rate of 0.5 ml per minute. Solvent A: 0.1% formic acid (Fisher Chemical Optima LC/MS grade; A11750) in water (Fisher Chemical Optima LC/MS grade; W6-4); solvent B: 0.1% formic acid in acetonitrile (Fisher Chemical Optima LC/MS grade; A955-4). A/B gradient started at 1% B for three minutes after injection and increased linearly to 99% B at 20 minutes, followed by five minutes at 99% B, then back to 1% B over 0.1 minute and finally held at 1% B for an additional 2.9 minutes.

Mass spectrometer parameters: spray voltage, -3.0 kV / +3.5 kV; capillary temperature 380˚C; probe heater temperature 400˚C; sheath, auxiliary, and sweep gas, 60, 20, and 2 AU, respectively; S-Lens RF level, 50; resolution, 70,000 at m/z 200; AGC target, 3E6. Each sample was analyzed in negative (ESI−) and positive (ESI+) electrospray ionization modes with m/z range 70–1000. Parameters for MS/MS (dd-MS2): MS1 resolution, 70,000; AGC Target, 1E6. MS2 resolution, 17,500; AGC Target, 2E5. Maximum injection time, 60 msec; Isolation window, 1.0 m/z; stepped normalized collision energy (NCE) 10, 30; dynamic exclusion, 1.5 sec; top five masses selected for MS/MS per scan. Peak areas were determined using Xcalibur Qual Browser (v4.1.31.9 Thermo Scientific) using a 5-ppm window around the *m/z* of interest.

## Yeast expression of mutant CYP-35D1

Site-directed mutagenesis (Q5 Site Mutagenesis Kit, New England Biolabs, Ipswich, MA, USA) was used to create a glutamate variant at position 267 of the codon-optimized cDNA

encoding *C. elegans* CYP-35D1. The plasmid carrying the constructed variant *C. elegans cyp-35D1* gene was digested at the flanking restriction enzyme sites *Spe*I and *Hind*III. The digest was subsequently run on a 1% agarose gel for band excision and gel purification. The insert was then ligated into the ATCC p416 GAL1 yeast expression vector (URA3, AmpR selection markers; CEN6/ARSH4 origin of replication [53] and transformed into competent DH5a *E. coli* cells. Plasmid DNA was purified from a single colony for sequence verification and subsequent transformation into *S. cerevisiae* BY4741 (*MATa his3Δ1 leu2Δ0 met15Δ0 ura3Δ0 pdr5Δ::hCPR_LEU2 snq2Δ::hb5_SpHIS5*). Transformants were selected on SD-URA agar plates.

## LC-MS/QTOF of yeast Lysates

**Yeast incubation.** P450-expressing yeast strains (wild-type and mutant (K267E)) as well as empty vector (EV) were incubated overnight in SD-URA selective medium with addition of 2% galactose with shaking on a rotating wheel at 37˚C. $OD_{600}$ was measured, and all strains were diluted to $OD_{600}$ = 10 in a final volume of 495 µL using the same selective medium. 5 µL of either DMSO (as a control) or 10 mM TBZ in DMSO was then added to each tube, for final concentrations of 1% DMSO or 100 µM TBZ, respectively, with subsequent incubation for six hours at 37˚C on a rotating wheel. Samples were prepared in technical triplicates, and each replicate was analyzed independently three times, unless otherwise noted. After the incubation, cells were filter-separated using Pall AcroPrep Advance 96-well filter plates (0.45 µm wwPTFE membrane, 1 ml well volume) on a vacuum manifold. Cells were then resuspended from the filter using 50 µL of autoclaved MilliQ water and frozen at -80˚C in 1.5 mL microtubes (Sarstedt P/N 72.694.006) before being used for the lysis.

**Yeast Lysis.** Cell pellets were thawed before starting the extraction. 400 µL of ACS grade 1-butanol and 0.5 mm glass beads (BioSpec Products P/N 11079105) were added to all samples and vortexed briefly. Samples were then homogenized using BioSpec Mini-Beadbeater-16 homogenizer at 3450 oscillations per minute in cycles of 30 seconds on and 30 seconds off for six cycles in total. Samples were then centrifuged for 10 minutes at 14000 rpm. The supernatant was transferred to 1.5 mL Eppendorf Safe-Lock tubes (Cat. No. 022363204). 400 µL of LC-MS grade methanol was added to the original tube with glass beads and homogenized the same way as with butanol. The samples were centrifuged again at the same speed and then transferred to the same Eppendorf Safe-Lock tube with butanol supernatant. Combined supernatants were dried at Eppendorf Vacufuge vacuum concentrator with a cold trap overnight at 30˚C.

**LC-MS/QTOF analysis.** Dried samples were resuspended using 100 µL of 50:50 acetonitrile:water (LC-MS grade solvents) with brief vortexing and then sonicated in a water bath for 15 minutes, followed by centrifugation at 20817 rcf for 10 minutes. 25 µL of each sample was then transferred to the polypropylene inserts (Agilent P/N 5182–0549) in amber vials immediately before the LC-MS/QTOF run. QTOF analysis increases the specificity of LC-MS by including a quadrupole (Q) filter that increases specificity by only allowing ions with a specific mass-to-charge ratio to pass through and then uses time-of-flight (TOF) as a measure of ion size. Samples were analyzed using the Agilent 1260 Infinity II with 6545 LC/QTOF mass spectrometer in positive ionization mode with Dual AJS electrospray ionization (ESI) equipped with Agilent ZORBAX Eclipse Plus C18 column (2.1x50mm, 1.8-µm particles) and ZORBAX Eclipse Plus C18 guard column (2.1x5mm, 1.8-µm particles). LC parameters used: injection volume 5µL with 10µL needle wash with sample, autosampler chamber temperature 4˚C, column oven temperature 40˚C. Mass spectrometry parameters used: gas temperature 320˚C, drying gas flow eight liters per minute, nebulizer 35 psi, sheath gas 350˚C at 11 liters per minute, VCap 3500V, Nozzle voltage 1000V, fragmentor 175V, skimmer 65V. The solvent gradient

with the flow of 0.5 ml per minute started with 99% mobile phase A (Optima LC/MS $H_2O$ +0.1% Formic Acid, Fisher Chemical P/N LS118-4) and 1% mobile phase B (Optima LC/MS Acetonitrile+0.1% Formic Acid, Fisher Chemical P/N LS120-4), kept for three minutes, increased linearly to 99% B at 20 minutes, followed by five minutes at 99% B, then back to 1% B over 0.1 min and finally held at 1% B for an additional 2.9 minutes. The post-run time was three minutes (instrument conditioning at 99% mobile phase A). The raw data was analyzed using Agilent MassHunter Qualitative Analysis 10.0. Counts of molecules with mass-to-charge (m/z) ratios specific to TBZ and TBZ-OH were collected, and the area under the curve of each peak was calculated to determine the abundance of each molecule. Abundance of TBZ-OH relative to the abundance of TBZ was calculated to quantify the enzyme effect. Genetic backgrounds of yeast strains expressing *C. elegans cyp-35D1* were identical, so differences in metabolism were attributed to the specific version of *cyp-35D1* expressed.

## Generation of *cyp-35D1* QTL region tree

We gathered genotype data for the QTL region (V:15734606–16365245) for strains present in the GWAS from the 20220216 VCF release from CaeNDR [23,24]. A tree was generated using the MUSCLE algorithm with default parameters [54]. Each point, representing an individual strain, in the tree was colored based on the allele at the 267 amino-acid position.

## Collection of orthologous sequences of *cyp-35D1*

Orthologs of *cyp-35D1* were identified in the three parental *C. elegans* strains used in this study (N2, CB4856, and DL238) and from representative strains from multiple supergroups within the *Caenorhabditis* genus using BLAST and the WormBase database [55–57]. *Caenorhabditis tropicalis*, *Caenorhabditis briggsae*, and *Caenorhabditis remanei* represent different clades within the *elegans* supergroup [55]. We used *Caenorhabditis becei* and *Caenorhabditis panamensis* as the representatives for the *japonica* supergroup. Outside of the *elegans* and *japonica* supergroups, we chose *Caenorhabditis bovis* as a distantly related *Caenorhabditis* species. The four closest orthologs in the parasite *H. contortus* were also included. *Diploscapter coronatus* was included as an outgroup for our tree construction. A tree was generated using VCF-kit [58]. An additional analysis of all predicted nematode orthologs of CYP-35D1 was also performed. All relevant WormBase accessions for *cyp-35D1* (and orthologs) sequences used can be found in S19 File.

## Analysis of *cyp-35D1* orthologs in *Haemonchus contortus*

The four most closely related *cyp-35D1* orthologs from *Haemonchus contortus* were analyzed for nonsynonymous mutations at the locus of interest. Amplicon sequencing was performed for each ortholog on 128 archived *H. contortus* samples collected from farms in the USA, Canada, and the UK that have been exposed to high levels of fenbendazole selection (S15 File). In addition, Fecal Egg Count Reduction Trials (FECRT) have demonstrated fenbendazole resistance in 22 of these populations of animals. A couple of laboratory *H. contortus* BZ-resistant strains (MHco18 and MHco10) and a laboratory *H. contortus* BZ-sensitive strain (MHco3ISE) were also tested. Primers were designed to amplify a product of approximately 350 bp, spanning the region containing codon 267. We created adapted primers suitable for Illumina next-generation sequencing and prepared amplicons for sequencing using a standard two-step PCR approach [15]. Illumina barcode indices, as well as the P5/P7 sequencing regions were added to the amplicons from each sample using a second (limited cycle) PCR to allow the pooling of up to 384 different samples in a single Illumina MiSeq library. Four separate pooled libraries, one for each ortholog, were sequenced using a 500 bp paired-end reagent kit (MiSeq Reagent

Kit v2, MS-103-2003) on an Illumina Desktop sequencer at a final concentration of 12 pM with the addition of 20% PhiX control v3 (Illumina, FC-110-3001). The raw sequencing reads were passed through a pipeline based on the analysis package DADA2 [59]. In brief, immediately following sequencing, raw data were demultiplexed, and the barcode indices were removed, resulting in the generation of FASTQ files for each sample. In turn, the pipeline removes primers from the sequence reads using the program Cutadapt [60] and then filters the reads based on size (>200 bp) and quality, using the filterAndTrim function to discard reads with a maximum of two expected errors in the forward read or five in the reverse read. DADA2 was used to generate error models and remove sequencing errors from raw reads [59]. Forward reads were then merged with the reverse reads, and possible chimeric sequences were removed from the dataset. Although the total number of samples sequenced was 128, data yield for analysis ranged between 103 and 118 samples for each ortholog. The resulting Amplified Sequence Variants (ASVs) were compared to the appropriate *H. contortus* reference sequence, and the frequency distribution of variation at position 267 in *H. contortus* ASVs across individual samples was generated.

### Tajima's D calculations

We calculated Tajima's D [61] 25 kb upstream and 25 kb downstream of *cyp-35D1* and *nhr-176* (V:16044238–16094238) using the *scikit-allel* package [62]. We calculated Tajima's D genome-wide based on a 10 kb window with a 1 kb sliding window.

### Statistical analysis

All statistical comparisons were performed in R (4.1.2) [63]. We used the *Rstatix* package *tukeyHSD* function on an ANOVA model generated with the formula *phenotype ~ strain* to calculate differences in the responses of the strains.

## Supporting information

**S1 File. Strains used in the manuscript.**
(TSV)

**S2 File. Processed phenotype data from GWAS.**
(TSV)

**S3 File. Processed mapping data from GWAS.**
(TSV)

**S4 File. Processed linkage mapping data.**
(TSV)

**S5 File. Processed *ben-1* regressed phenotype data from GWAS.**
(TSV)

**S6 File. Processed *ben-1* mapping data from GWAS.**
(TSV)

**S7 File. Processed data for peak marker in GWAS and linkage mapping experiments.**
(TSV)

**S8 File. Fine mapping data for the peak marker on chromosome V.**
(ZIP)

**S9 File. Processed HTA data for *cyp-35D1* deletions and K267D mutants.**
(CSV)

**S10 File. Processed HTA data for *cyp-35D1* and *nhr-176* deletion comparisons.**
(CSV)

**S11 File. Processed HTA data for NILs and allele swap strains.**
(CSV)

**S12 File. Processed allele frequency data for competition assays.**
(CSV)

**S13 File. Processed relative fitness data for competition assays.**
(CSV)

**S14 File. Processed metabolomics data.**
(XLSX)

**S15 File. Processed yeast metabolomics data.**
(CSV)

**S16 File. Details of *H. contortus* populations and full results of amplicon sequencing.**
(XLSX)

**S17 File. Tajima's D values around the *nhr-176 cyp-35D1* locus.**
(CSV)

**S18 File. Oligonucleotide sequences used in the manuscript.**
(TSV)

**S19 File. Sequence accession information for all sequences used in analyses.**
(TSV)

**S1 Fig. Regression of *ben-1* variation emphasizes a prominent QTL on chromosome V.** (A) Distribution of normalized TBZ response after regression using *ben-1* variation as a covariate is shown ordered from most susceptible to most resistant. The N2 and CB4856 strains are colored orange and blue, respectively. Strains with variation in *ben-1* have a red triangle at the base of the bar for that strain. (B) Genome-wide association mapping results for animal length following *ben-1* regression are shown. The genomic position is shown on the x-axis, and -log10($p$) values are shown on the y-axis for each SNV. SNVs are colored pink if they pass the Eigen threshold (dashed horizontal line) or red if they pass the Bonferroni significance threshold (horizontal line).
(TIF)

**S2 Fig. Phenotype by genotype plots for peak marker in genome-wide association mapping and linkage mapping confirm greater susceptibility in CB4856.** (A) Regressed animal length (mean.TOF) values in response to TBZ treatment in a genome-wide association study are shown on the y-axis. The x-axis denotes if a strain has the lysine (REF) or glutamate (ALT) alleles at the peak marker. Each point represents a strain's normalized response from multiple replicates. (B) For the QTL discovered in the linkage mapping experiment, the regressed animal length (mean.TOF) is shown on the y-axis, and the allele of the tested recombinant strain is shown on the x-axis. Data are shown as box plots with the median as a solid horizontal line, the top and bottom of the box representing the 75th and 25th quartiles, respectively.
(TIF)

**S3 Fig. NILs confirm region on chromosome V underlies TBZ response.** The strain tested is shown on the y-axis. (A) Chromosome V of each strain with the genotype at markers across chromosome V is shown. The x-axis shows the genomic position across the chromosome. A red line denotes the location of the QTL on chromosome V. The genomic background of the *cyp-35D1* locus is shown orange for N2 and blue for CB4856. (C) Regressed animal length (mean.TOF) in response to TBZ treatment is shown on the x-axis. The ECA238 strain is a near-isogenic line in the N2 genetic background with a small introgression of the CB4856 genome around the identified chromosome V QTL from linkage mapping. The ECA239 strain is a near-isogenic line in the CB4856 background with a small introgression of the N2 genome around the identified QTL from linkage mapping. Data are shown as Tukey box plots with the median as a solid vertical line, the right and left of the box representing the 75th and 25th quartiles, respectively. Statistical significance in comparison to the genomic background strain is shown to the right ($p < 0.0001$ = ****, Tukey HSD).
(TIF)

**S4 Fig. Fine mapping of the QTL region indicates that *cyp-35D1* is correlated with response to TBZ in non-*ben-1* regressed data and more so after regression.** Fine mapping of the QTL region on chromosome V is displayed for (A) non-*ben-1*-regressed data. Each gray bar represents a gene in the region of interest. Red bars indicate high-impact variants in the region, black arrows indicate the direction of gene transcription, and the association between TBZ response and the variant is shown on the y-axis. The location of *cyp-35D1* is shown with a label and an arrow pointing to the gene and variant location
(TIF)

**S5 Fig. Deletion of *cyp-35D1* confers greater levels of susceptibility than E267, and deletion of *nhr-176* confers even greater levels of susceptibility.** The strain *cyp-35D1* genotype (top) and *nhr-176* genotype (bottom) are shown on the x-axis, with WT representing the wild type for each gene. N2 (orange) and CB4856 (blue) are shown with comparisons to strains within the same genetic background. Data from both independent edits for each edit are shown. Regressed median animal length values of response to TBZ are shown on the y-axis Each point represents a well that contains ~30 animals after 48 hours of exposure to TBZ. Data are shown as box plots with the median as a solid horizontal line, the top and bottom of the box representing the 75th and 25th quartiles, respectively. The top and bottom whiskers are extended to the maximum point that is within 1.5 interquartile range from the 75th and 25th quartiles, respectively. Statistical significance in comparison to the genomic background strain is shown above each strain ($p < 0.0001$ = ****, Tukey HSD).
(TIF)

**S6 Fig. Deletion of *cyp-35D1* and *nhr-176* confers susceptibility equivalent to deletion of *nhr-176* alone.** The strain *cyp-35D1* genotype (top) and *nhr-176* genotype (bottom) are shown on the x-axis with WT representing the wild type for each gene. N2 (orange) and CB4856 (blue) are shown for comparisons with genetic backgrounds. Regressed median animal length values of response to TBZ are shown on the y-axis Each point represents a well that contains ~30 animals after 48 hours of exposure to TBZ. Data are shown as box plots with the median as a solid horizontal line, the top and bottom of the box representing the 75th and 25th quartiles, respectively. The top and bottom whiskers are extended to the maximum point that is within 1.5 interquartile range from the 75th and 25th quartiles, respectively. Statistical significance in comparison to the genomic background strain is shown above each strain ($p < 0.0001$ = ****, Tukey HSD).
(TIF)

**S7 Fig. Amino acid substitution at codon 267 of *cyp-35D1* confers increased susceptibility to TBZ in CRISPR-Cas9 edited strains.** The strain name is displayed on the y-axis. (A) The genomic background of the tested strain is shown as orange or blue for N2 and blue CB4856, respectively. The CYP-35D1 allele for the strain tested is shown as orange for K267 and blue for E267. (C) Regressed animal length (mean.TOF) values of response to TBZ are shown on the x-axis. Each point represents a well that contains hundreds of animals following 96 hours of TBZ treatment. Data are shown as box plots with the median as a solid vertical line, the right and left of the box representing the 75th and 25th quartiles, respectively. The left and right whiskers are extended to the maximum point that is within 1.5 interquartile range from the 25th and 75th quartiles, respectively. Statistical significance in comparison to the genomic background strain is shown to the right of each strain; N2 and CB4856 are also significantly different (p<0.05 = *, $p < 0.0001$ = ****, Tukey HSD).
(TIF)

**S8 Fig. Response of alleles at position 267 indicates that animals with aspartate are the most susceptible, followed by glutamate.** Regressed mean animal length values of responses to TBZ, both non-regressed by *ben-1* (A) and regressed by *ben-1* (B), are shown on the y-axis. Each point represents the regressed mean animal length value from hundreds of animals in a single well. The x-axis shows the CYP-35D1 allele. Data are shown as box plots with the median as a solid horizontal line, the top and bottom of the box representing the 75th and 25th quartiles, respectively. The top and bottom whiskers are extended to the maximum point that is within 1.5 interquartile range from the 75th and 25th quartiles, respectively.
(TIF)

**S9 Fig. Aspartate substitution as position 267 does not confer susceptibility.** The strain name is displayed on the y-axis. Two independent edits for each allele change are shown. (A) The genomic background of the tested strain is shown as orange or cadet-blue for N2 and DL238, respectively. Alleles of CYP-35D1 are shown in the same color as the relevant parental genotype, and strains with an N2 background but with the glutamate allele are shown in blue. (B) Regressed median animal length values of response to TBZ are shown on the x-axis Each point represents a well that contains ~30 animals after 48 hours of exposure to TBZ. Data are shown as box plots with the median as a solid vertical line, the right and left of the box representing the 75th and 25th quartiles, respectively. The left and right whiskers are extended to the maximum point that is within 1.5 interquartile range from the 25th and 75th quartiles, respectively. Statistical significance in com parison to the genomic background strain is shown above each strain ($p > 0.05$ = ns, $p < 0.01$ = **, $p < 0.0001$ = ****, Tukey HSD).
(TIF)

**S10 Fig. The abundances of TBZ and TBZ metabolites in the endo-metabolomes two and six hours after exposure.** The change in the abundances of TBZ (A) and three metabolites: TBZ-OH (B), TBZ-O-glucoside (C), and TBZ-O-phosphoglucoside (D) are shown, with samples taken at two and six hours after exposure to 50 μM TBZ. CYP-35D1 alleles are shown on the x-axis with K267 (orange) and K267E (light gray) in the N2 genetic background, and E267 (blue) and E267K (dark gray) in the CB4856 genetic background. Abundances are shown as the log of abundance after normalization for the abundance of ascr#2. The line represents the mean abundance, and each point represents an individual replicate. Statistical significance between strains with the same genetic background at the same time point is shown ($p > 0.05$ = ns, $p < 0.05$ = *, $p < 0.0001$ = ****, Wilcoxon Rank Sum test with Bonferroni correction t-test of Independent Replicates).
(TIF)

**S11 Fig. The abundances of TBZ and TBZ metabolites in the exo-metabolomes two and six hours after exposure.** The change in the abundances of TBZ (A) and three metabolites: TBZ-OH (B), TBZ-O-glucoside (C), and TBZ-O-phosphoglucoside (D) are shown, with samples taken at two and six hours after exposure to 50 μM TBZ. CYP-35D1 alleles are shown on the x-axis with K267 (orange) and K267E (light gray) in the N2 genetic background, and E267 (blue) and E267K (dark gray) in the CB4856 genetic background. Abundances are shown as the log of abundance after normalization for the abundance of ascr#2. The line represents the mean abundance, and each point represents an individual replicate. Statistical significance between strains with the same genetic background at the same time point is shown ($p > 0.05$ = ns, Wilcoxon Rank Sum test with Bonferroni correction t-test of Independent Replicates).
(TIF)

**S12 Fig. Tajima's D around cyp-35D1 and nhr-176 locus.** The divergence measured by Tajima's D surrounds the cyp-35D1 and nhr-176 locus on chromosome V. The most recent CaeNDR variant release was used to calculate Tajima's D in this region [31,58].
(TIF)

## Acknowledgments

We would like to thank members of the Andersen lab for making reagents used in the experiments and their feedback in the preparation of this manuscript. We acknowledge the *Caenorhabditis* Natural Diversity Resource (NSF Capacity Grant 2224885) for providing the strains used in this study. Additionally, we would also like to acknowledge WormBase for providing an essential resource for genetic and genomic data used in this manuscript.

## Author Contributions

**Conceptualization:** J. B. Collins, Clayton M. Dilks, Erik C. Andersen.

**Data curation:** J. B. Collins, Clayton M. Dilks, Steffen R. Hahnel, Brittany Cooke, Kateryna Sihuta, Mostafa Zamanian.

**Formal analysis:** J. B. Collins, Clayton M. Dilks, Steffen R. Hahnel, Bennett W. Fox, Mostafa Zamanian.

**Funding acquisition:** Erik C. Andersen.

**Investigation:** J. B. Collins, Clayton M. Dilks, Steffen R. Hahnel, Briana Rodriguez, Bennett W. Fox, Elizabeth Redman, Jingfang Yu, Brittany Cooke, Kateryna Sihuta, Mostafa Zamanian, Peter J. Roy, Frank C. Schroeder, John S. Gilleard.

**Methodology:** J. B. Collins, Clayton M. Dilks, Steffen R. Hahnel, Bennett W. Fox, Mostafa Zamanian, John S. Gilleard, Erik C. Andersen.

**Project administration:** Erik C. Andersen.

**Resources:** Peter J. Roy, Frank C. Schroeder, Erik C. Andersen.

**Supervision:** Peter J. Roy, Frank C. Schroeder, John S. Gilleard, Erik C. Andersen.

**Writing – original draft:** J. B. Collins, Bennett W. Fox, Peter J. Roy, Erik C. Andersen.

**Writing – review & editing:** J. B. Collins, Steffen R. Hahnel, Bennett W. Fox, Elizabeth Redman, Mostafa Zamanian, Peter J. Roy, John S. Gilleard, Erik C. Andersen.

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
