## [Decision Letter · Decision Letter 0]

24 Oct 2024

PPATHOGENS-D-24-02017Naturally occurring variation in a cytochrome P450 modifies thiabendazole responses independent of beta-tubulinPLOS Pathogens Dear Dr. Andersen, Thank you for submitting your manuscript to PLOS Pathogens. After careful consideration, we feel that it has merit but does not fully meet PLOS Pathogens's publication criteria as it currently stands. Therefore, we invite you to submit a revised version of the manuscript that addresses the points raised during the review process. Please submit your revised manuscript within 60 days Dec 23 2024 11:59PM. If you will need more time than this to complete your revisions, please reply to this message or contact the journal office at plospathogens@plos.org. Please include the following items when submitting your revised manuscript:*
A rebuttal letter that responds to each point raised by the editor and reviewer(s). You should upload this letter as a separate file labeled 'Response to Reviewers'. This file does not need to include responses to any formatting updates and technical items listed in the 'Journal Requirements' section below.*
A marked-up copy of your manuscript that highlights changes made to the original version. You should upload this as a separate file labeled 'Revised Manuscript with Track Changes'.*
An unmarked version of your revised paper without tracked changes. You should upload this as a separate file labeled 'Manuscript'. If you would like to make changes to your financial disclosure, competing interests statement, or data availability statement, please make these updates within the submission form at the time of resubmission. Guidelines for resubmitting your figure files are available below the reviewer comments at the end of this letter. We look forward to receiving your revised manuscript. Kind regards, Adler R. Dillman, Ph.D.Academic EditorPLOS Pathogens James Collins IIISection EditorPLOS Pathogens Michael Malim

Editor-in-Chief

PLOS Pathogens

orcid.org/0000-0002-7699-2064   **Journal Requirements:** **Additional Editor Comments (if provided):** The reviewers appreciated the attention to an important problem, but raised some substantial concerns about the manuscript as it currently stands. Based upon my own reading of the manuscript and the consensus of the three reviewers, this is a very interesting and thorough study of a non-ben-1-based genetic polymorphism that modulates resistance to thiobendazole. The use of powerful tools to advance our understanding of the importance of this polymorphism in explaining resistance and susceptibility of C. elegans and find supporting metabolic evidence both in C. elegans and in yeast. Overall the experiments were very thorough and the progression of the manuscript is good. In the aggregate, the three reviewers outline several major issues that the authors will need to address, and I agree with these. It seems that the concerns and suggested revisions are intentionally limited in scope and feasible to address. The reviewer comments will help flesh out and improve a manuscript that is already of high quality. I look forward to seeing the revised version.**Reviewers' Comments:** Reviewer's Responses to Questions

**Part I - Summary**

Reviewer #1: In “Naturally occurring variation in a cytochrome P450 modifies thiabendazole responses independent of beta-tubulin”, Collins and colleagues describe their investigation of the basis of naturally occurring nematicide resistance in a large collection of wild isolates of the free-living soil nematode Caenorhabditis elegans. They report discovering a polymorphism in a cytochrome P450 gene that promotes the resistance of animals to thiabendazole and affects thiabendazole metabolism in C. elegans and when the protein is expressed in yeast. They seek evidence for whether an equivalent polymorphism might arise in isolates of the parasitic nematode Haemonchus contortus that have been exposed to a related nematicide.

This is an important field of study, and the authors have deployed powerful tools to advance our understanding of it. They thoroughly demonstrate the functional importance of this polymorphism in explaining resistance and susceptibility of the laboratory standard wild type strain and wild isolates of C. elegans, and find supporting metabolic evidence both in C. elegans and in yeast. The authors’ findings seem very appropriate for this journal.

I list below a large number of comments, questions, and suggestions for the authors to consider, both experimental and in describing their results. I realize that the comments achieve significant length, but I have tried to be restrained in proposing time-consuming experiments, and some experimental issues that I mention might reasonably be ignored at this time, or addressed only by considering them in writing. There are however some experimental issues that should be addressed for a basic understanding of the nematicide resistance polymorphism the authors have found, including whether the cyp-35D1 267K polymorphism provides resistance to nematicides related to thiabendazole; its effectiveness against different dosages of nematicide, and especially against dosages relevant to those used medically and agriculturally; and whether the resistance phenotype is dominant or recessive at relevant dosages.

Reviewer #2: Collins et al. have produced an important and high-quality analysis of a non-ben-1-based genetic polymorphism in Caenorhabditis elegans that modulates resistance to thiobendazole, a drug closely related to a key antihelmintic albendazole that is under heavy pressure from the rise of genetic resistance in Haemonchus contortus and other parasitic nematodes. I do not have any serious criticisms of this work; I only have some comments on the evolution of cyp-35d1 homologs, thoughts about possibly interesting implications of the authors' work, and some proposed minor textual revisions.

Reviewer #3: Collins et al. report a polymorphism impacting thiabendazole resistance in C. elegans. Notably, this allele impacts resistance independently of tubulin genes, which have been frequently implicated in anthelminthic resistance. They perform reciprocal gene editing experiments to show that the allele is causative, and they also perform MS to examine the impacts of this allele on thiabendazole metabolism. All together, this is a valuable study that should be of interest to the readership of this journal.

**Part II – Major Issues: Key Experiments Required for Acceptance**

Reviewer #1: 1) Dosage of thiabendazole (TBZ) used is not always clearly presented. The dosages used are never compared to dosages used medically or agriculturally or encountered environmentally, a question of obvious relevance that is never mentioned. No comparison is made between the level of TBZ resistance achieved through the cytochrome 267K polymorphism and that possible through mutation of the tubulin gene ben-1. A dose-response assay would seem ideal, comparing relevant cyp-35D1 genotypes to each other and potentially to ben-1 mutants, but at the least this should be discussed and should be considered more thoroughly.

2) At one point in the manuscript (circa line 434) the authors state that although they have used TBZ throughout this study, another related molecule fenbendazole (FBZ) is now in much greater use than TBZ. For example, the H. contortus isolates examined for cyp-35D1 homolog mutations had been treated not with TBZ but with FBZ. It would be good to show whether mutations of cyp-35D1 can affect susceptibility to FBZ in C. elegans. This also applies to albendazole (ABZ), which is only mentioned once in the manuscript to cite a reference.

3) The authors demonstrate that under laboratory conditions, over a few generations, there might not be any deleterious or advantageous effect of the cyp-35D1 267K polymorphism in competition assays. This result could be very important, but the data indicating it seem to be limited. Laboratory culture conditions are not highly selective. Furthermore, it isn’t clear from the authors’ presentation of the data (Figure 3A) whether there is indeed no difference between the 267E and 267K alleles; the authors report no statistically significant difference, but there appears possibly to be both increased variation and a downward trend from the 267E allele. It is possible that an increased number of trials or an increase in number of generations would result in this trend achieving statistical significance – which could in turn explain why the 267K polymorphism became fixed over many generations of culturing C. elegans under these same laboratory conditions prior to the strain first being frozen. The authors also do not report whether there is any effect of cyp-35D1 deletion alleles, either in competition assays or by measuring brood size and time to maturation. Similarly the authors do not discuss whether there are any reported effects on health or fecundity of TBZ resistance mutations in ben-1.

4) The authors report looking for cytochrome P450 changes corresponding to the 267K polymorphism in the genomes of selected Caenorhabditis nematodes and by sequencing samples from 128 samples of FBZ-exposed populations of the parasitic nematode H. contortus. In none of these do they report finding a 267K polymorphism, including none arising in the FBZ-exposed H. contortus (in which context it seems again relevant that the authors do not report having tested whether C. elegans with the 267K genotype are resistant to FBZ). The authors should look more systematically at nematode genomes, and should more thoroughly describe the cytochrome P450 phylogeny in nematodes. An admittedly sloppy survey of Caenorhabditis finds that the first BLAST hit when querying with C. elegans CYP-35D1 often – perhaps half of the time – finds an asparagine at the position corresponding to amino acid 267 of CYP-35D1. One of the four H. contortus homologs of CYP-35D1 that the authors describe also has this 267N sequence, which the authors note but do not discuss. It’s possible these homologous proteins (both the Caenorhabditis homologs with 267N and the H. contortus protein with 267N) are orthologous to cytochromes of C. elegans other than CYP-35D1 (there is at least one C. elegans cytochrome with an asparagine at this position), but the authors should discuss this. Moreover, 267K sequence can be found in at least two other nematode species (the pine tree nematode Bursaphelenchus okinawaensis and the mammalian parasite Strongyloides stercoralis), and still other amino acids can be found in this position in highly ranked BLAST hits from other available nematode genomes (Q in Deladenus siricidicola, Nippostrongylus brasiliensis, Poikilolaimus oxycercus, and Steinernema carpocapsae; also G in Steinernema carpocapsae; and P in Panagrellus redivivus). It is striking that the authors experimentally investigated the 267D polymorphism found in one isolate of C. elegans (a highly conservative change from 267E that can be found in many homologs of CYP-35D1 throughout nematodes) but do not discuss any other variants, not even the asparagine variant they report having found in H. contortus.

5) nhr-176 is mentioned extremely briefly in the manuscript, is squarely in the region implicated by genetic mapping, and according to the literature described by the authors is directly relevant to TBZ resistance, but almost as soon as it is mentioned nhr-176 is discarded from the manuscript, a choice that is never fully explained. This is particularly striking on line 110 (“We further narrowed this region to two candidate genes including the cytochrome P450 [gene] cyp-35d1”). The other gene in that region is nhr-176, which is not mentioned. At no point do the authors clearly state that nhr-176 is immediately adjacent to cyp-35D1 and even appears at a glance to be co-transcribed with cyp-35D1 in a single operon.

6) Speaking of the possibility that nhr-176 and cyp-35D1 might be in an operon: the authors report generating deletion mutants of each gene, but do not report whether either deletion allele impairs the other gene, either by looking for the expression of the mRNA or by using complementation to test the function of the other gene. The authors mutate both genes separately in cis, without first demonstrating whether they had already functionally done so by disrupting the potential operon.

7) The authors do not report testing whether TBZ resistance caused by the 267K polymorphism of cyp-35D1 is dominant or recessive, at any dosage of TBZ, nor do they state what is known about dominant or recessive action of ben-1 mutations. This seems of obvious importance, especially for parasitic nematode species that might homozygose newly emerging mutations less quickly than does the selfing hermaphrodite C. elegans.

8) The authors report different levels of activity when different versions of CYP-35D1 are expressed in yeast from identical plasmids. They don’t report doing anything to confirm identical levels of protein expression or stability in their transgenic yeast. I believe the plasmids they used should avoid issues of plasmid copy number, but even so the reader should be reassured that these potential caveats have been addressed experimentally or at least that they have been considered and discarded according to the standard of the field.

Reviewer #2: I see no new key experiments or modifications of existing experiments that should be required for acceptance.

Reviewer #3: (No Response)

**Part III – Minor Issues: Editorial and Data Presentation Modifications**

Reviewer #1: 9) According to WormBase the C. elegans gene name is “cyp-35D1”, not (as it consistently appears in the manuscript) “cyp-35d1”. I believe the H. contortus gene homologous to C. elegans ben-1 may be “Hco-tbb-iso-1” (it appears in the literature in that way at least once), but it surely cannot be “Hco-isotype-1”, as it appears in this manuscript at least twice.

10) Colors are used in histograms in the Figures that are frequently not mentioned in the legends and are not obviously consistent from figure to figure.

11) Line 83: It is striking that the introduction considers parasitic nematodes and their relevance to human and veterinary health, but does not mention plant-parasitic nematodes and their control in agriculture. This might easily be relevant to the emergence of TBZ-resistant phenotypes in free-living soil nematodes such as C. elegans, and also to the presence of a polymorphism equivalent to cyp-35D1 267K in the plant-feeding nematode Bursaphelenchus okinawaensis. The authors similarly do not mention the use of TBZ and related drugs against parasitic invertebrates other than nematodes.

12) Line 142: The text says “Highly TBZ-resistant strains had predicted loss-of-function variation in ben-1 (Fig 1A)”. Every strain in Figure 1A marked with a red triangle (defined as “with variation in ben-1”) demonstrates increased sensitivity to ben-1, not increased resistance. This is confusing.

13) Line 148: List the six beta tubulin genes, here or in the Methods.

14) Line 152: The text here attributes to reference 19 a distinction between the QTLs on LGII and on LGV in their interactions with the ben-1 locus. This distinction is not obviously made in reference 19, which states that controlling for variation in ben-1 “resulted in the disappearance of the two QTL on chromosome II and the QTL on chromosome V”. Please clarify.

15) Line 183 (Figure 1): The label in the figure “Neither” could be clearer. The legend should state that 214 strains are shown.

16) Line 272: The manuscript states that introduction of the 267E polymorphism to the N2 strain did not appear to reduce TBZ metabolism, even though the corresponding and opposite change in CB4856 increased metabolism. This is briefly stated and is never discussed.

17) Line 277: “yeast” should be “Saccharomyces cerevisiae”. Also, a glance at the literature indicates that expression of metazoan cytochromes in yeast to test their metabolic function is a standard method, but this is not made obvious in the manuscript, nor is it mentioned why it’s seen as an appropriate system – and unless I have overlooked it none of this literature seems to have been cited in the Results, the Discussion, or the Methods.

18) Line 285 (Figure 4): Four asterisks is not defined in the legend. Strain backgrounds could be more clearly labeled, rather than being inferred from starting genotype.

19) Line 318: Should “area” be plural? I am confused by this statement.

20) Line 320: This dendrogram is offered to the reader, but is never explained to them. Does it indicate a single, relatively recent origin of the 267K polymorphism, that has been spread rapidly? How much more information or at least argumentation can be extracted from this dendrogram?

21) Lines 322-325: This is an argument to be made in the Discussion, not the Results. Also, given that the 267K polymorphism appears to be at worst neutral, the argument should be made at greater length, potentially citing relevant work.

22) Line 328 (Figure 6): In designing figures the authors should consider the incidence of color-blindness. Also, as mentioned previously, extremely little effort is made to explain the figure to the reader or to discuss its significance. This is frequent throughout the manuscript.

23) Line 398: The authors propose that the TBZ-resistant cyp-35D1 polymorphism might have arisen in the strain N2 and become common in isolates more recently found around the world because of the presence in the environment of human-used TBZ and related drugs, and also naturally occurring related molecules. Please clarify when use of TBZ or related molecules began, and whether this could explain why a C. elegans isolate obtained in Bristol more than five decades ago and subsequently frozen has this TBZ resistance polymorphism.

24) Line 423: I assume “predicted high impact” means a nonconservative coding change?

25) Line 424: “although we can draw no conclusions about the resistance status”. This seems like it could be overcome, if this were desired.

26) Line 434: Expand on the statement about the discontinued use of TBZ. Why was it stopped? When was it stopped? Is the cyp-35D1 267K polymorphism specific to TBZ, versus other related drugs?

27) Line 465: This section is unclear. Were 219 new RIAILs generated for use in this study, according to the methods described in reference 22, or is this study referring to a subset of the 359 RIAILs generated in that referenced work? Were the sequence reads deposited in the Short Read Archive?

28) Line 471: Despite the statement that all strains are listed in Supplementary File 1 and are available from the CaeNDR, Supplementary File 1 does not list 219 RIAILs, does not seem to obviously list any RIAILs. Were the RIAILs generated for this study, or not? If so, were they all preserved? Were any preserved?

29) Line 471: Supplementary File 1 does not appear to list the cyp-35D1 nhr-176 double mutants used in Figures S5 and S6.

30) Line 473: This paragraph does not mention the generation of the double deletion mutants used in Figures S5 and S6.

31) Line 473: Mention here in the Methods that oligonucleotide sequences are in Supplementary File 1.

32) Line 473: Oligos listed in Supplementary File 1 include the “First crispr guide for cyp-35D1 deletion” but not any second guide, nor any oligonucleotide or construct used to guide repair, nor do they list any reagents for the nhr-176 deletions.

33) Line 477: The sequence changes in the deletion alleles are not described (and as mentioned above the sequences of the reagents used to generate them are not provided).

34) Line 485: Give or cite a detailed protocol for “bleach synchronized”.

35) Lines 488 and 490: “L1” and “L4” are not defined.

36) Line 628: Name, cite, and/or acknowledge the codon optimization tool used. Provide codon-optimized sequences in the Supplementary Files or upon request. Provide more information about site-directed mutagenesis: cite a reference or name a kit, at least. Note that any oligonucleotides used in this section are not included in Supplementary File 1.

37) Line 693: Provide accession numbers or other identifying information for the sequences used.

38) Line 746: WormBase should be cited according to the FAQ on their website.

39) Line 983 (Figure S2): Legend should do more to explain the figure to the reader.

40) Line 1009 (Figure S4): Black angle-bracket shapes (gene models maybe?) are not mentioned in the legend, and nhr-176 is not marked in the figure.

41) Line 1016 (Figure S5): Top and bottom rows below the figure should be labeled in the figure, not merely described in the legend. “WT” should be defined in the legend. Allele numbers should be stated, not merely “del”, considering that the whole point appears to be that two different deletion alleles were used. The 2nd, 3rd, 7th, and 8th rows all have equivalent genotypes, but the reason for this, and for the comparisons being made, is not clear. Yellow and blue colors in histograms not explained.

42) Line 1044 (Figure S6): Comments to Figure S5 apply.

43) Line 1061 (Figure S7): Single asterisk not defined in legend. Comparisons are not “above each strain”. Strain names should be labeled as strain names in the figure. Legend should clarify whether these are introgression strains or CRISPR strains.

44) Line 1095 (Figure S9): Dark blue not defined in legend with other colors. Typo in figure heading. Figure legend refers to a part C not in evidence. Parts A and B are not separate from each other.

45) Line 1108 (Figure S10): Legend defines a two-asterisk condition not present in the manuscript.

46) Line 1132 (Figure S12): I do not think this figure is mentioned in the manuscript.

Reviewer #2: EVOLUTION OF CYP-35D1

Lines 340-342: The authors write, "The relative distance between the elegans and japonica groups suggests that the elegans group cyp-35d1 has recently evolved and is not closely related to cyp genes in other Caenorhabditis species." I think this is probably a mistaken interpretation that arises from a poorly-publicized defect of the C. japonica genome that is still the one official C. japonica genome in ParaSite (https://parasite.wormbase.org), for reasons explained immediately below.

Figure 7: The authors intelligently picked both C. japonica and C. panamensis as representatives of the japonica supergroup (as described in their Methods). Their phylogeny shows that C. panamensis has a cyp-35d1 ortholog that falls phylogenetically within the C. bovis outgroup ortholog. However, their phylogeny has one putative C. japonica ortholog that falls completely outside the otherwise monophyletic Caenorhabditis clade in their Figure 7A phylogeny.

I think the reason they got this strange result is that the C. japonica genome, unlike almost every other Caenorhabditis genome assembly that was produced from 2010 onward, is incomplete (despite the fact that it was assembled early and expensively by the genome center at Washington University). This fact can be easily observed by going to the species genome page at ParaSite (https://parasite.wormbase.org/species.html) and looking at BUSCO scores for the 22 Caenorhabditis species' genomes on that page. There are 18 Caenorhabditis species' genome sequences that have genome BUSCO completeness scores of 97.0% to 98.6%. In contrast, the C. japonica genome assembly has a genome BUSCO completeness score of 92.4% (whereas the equivalent score for C. panamensis is 97.8%). I therefore suspect that in real life, C. japonica has a much closer ortholog of cyp-35d1 that on a phylogenetic tree would cluster very closely with the ortholog from C. panamensis. There does exist a (so far unpublished) superior C. japonica genome assembly by Lewis et al. (https://www.ncbi.nlm.nih.gov/datasets/genome/GCA_963572235.1) that the authors might search for a better cyp-35d1 ortholog (though they would need to manually annotate such a gene; the new and improved C. japonica genome is so far unannotated with protein-coding genes). Alternatively, the authors might replace C. japonica with some other japonica supergroup species with a better-assembled and annotated genome, and recompute their phylogeny and multiple sequence alignment in Figure 7 using a japonica-supergroup replacement for C. japonica. However, I do think that the authors should not continue taking their results with C. japonica at face value, and making statements about the evolution of this gene based on what may be only an apparently high divergence of this gene in C. japonica.

POSSIBLY INTERESTING IMPLICATIONS

1. The authors show a tight association between two adjacent genes in C. elegans, cyp-35d1 and nhr-176. Examining WormBase (https://wormbase.org) for their expression data shows that both of these genes are strongly expressed in the intestine. Surai and Hobert ([2021], PubMed 34348120) showed that many nhr genes in C. elegans appear to be expressed in neurons and may act to amplify sensory responses by binding to small molecules and then upregulating seven-transmembrane receptors for those small molecules. The authors' work on nhr-176 shows an interesting counterexample in which an NHR protein is not used for sensory responses in neurons but for detoxification responses in the intestine. This may be a more general phenomenon in nematodes.

2. The authors scanned many genomes of fenbendazole-resistant Haemonchus contortus isolates for lysine alleles in four H. contortus orthologs of cyp-35d1. I can confirm that orthology analysis identifies precisely these four genes, and that one of these four genes (HCON_00022670) has strongly intestine-biased RNA-seq expression (as computed from published data by Laing et al. [2013], PubMed 23985316). However, in addition to these four strict orthologs, H. contortus' genome also encodes 21 other cyp genes (defined by presence of the Pfam motif p450 [PF00067.25]), of which four show intestinal-biased gene expression. I find that three of them have more strongly intestinally-biased expression than HCON_00022670.

Gene FC FDR

HCON_00141025 8.10319755 3.11e-23

HCON_00084620 3.549094835 1.22e-12

HCON_00143950 3.39756099 1.21e-19

HCON_00022670 2.974729846 6.90e-13 # cyp-35d1 ortholog

HCON_00024010 2.678863 2.98e-14

FC == Hco_fem_gut vs. Hco_fem_worm log2FC

FDR == Hco_fem_gut vs. Hco_fem_worm FDR

It is outside the scope of this paper, but in future work, the authors may wish to scan these other H. contortus cyp genes with strongly intestinal expression for lysine alleles in fenbendazole-resistant isolates, since the selective presssure of albendazole resistance may lead to mutations in intestinal nhr genes that are not strictly orthologous to cyp-35d1. In their Discussion, the authors may wish to note the possibility that resistance alleles may arise in H. contortus cyp genes that are not strict cyp-35d1 orthologs but do have strong intestinally expression and thus are possible detoxification enzymes.

PROPOSED MINOR TEXTUAL REVISIONS

Title --

Line 2: "independent" should read "independently" (since it is an adverb modifying the verb 'modifies').

Abstract --

Line 55: "identified by another mapping technique" might be better written more specifically; maybe it could be something like "identified by linkage and fine mapping"?

Author summary --

Line 75: "in genes independent of beta-tubulin" might be better written as "in additional genes other than beta-tubulin".

Introduction --

Lines 99-100: "Using the cycle of discovery" might be better written as "Using this cycle of discovery".

Line 105: "populations and research" might be better written as "populations, and research".

Line 106: "independent" should read "independently".

Lines 118-119: Might "and was significantly favored" be better written as "but was significantly favored"?

Results --

Figure 1A: The label on the Y-axis ("TBZ Response") is confusing. Would it be acceptable to relabel it as "TBZ Resistance", which is what I think it really shows?

Line 191: What is an "Eigen significance threshold"? How is it defined? [This question also arises at Line 980 of the Supplementary Figures.]

Figure 2 and Line 229: There is inconsistent labeling of a lower significance threshold; it is labeled with two asterisks ('**') in the figure itself, but apparently with three asterisks ('***') in the figure legend. This inconsistency should be corrected.

Figure 4 and Lines 297-298: A single asterisk ('*') is labeled as being p < 0.05, but there is no explanation given for what four asterisks ('****') mean; this should be corrected.

Figure 5 and Lines 310-311: A single asterisk ('*') is labeled as being p < 0.05, but does not exist anywhere in the figure; the figure does have four asterisks ('****'), but there is no explanation given for what they mean. This should be corrected.

Line 392: "the cytochrome P450" might better read "a cytochrome P450".

Line 393: "independent" should read "independently".

Line 704: "Haemoncus" should read "Haemonchus".

Supplementary Figures --

Line 980: Again, what is an Eigen threshold and how is it defined?

Figure S7 and Lines 1075-1076: There is a significance of one asterisk ('*') shown in the figure, but not defined in the figure legend; this should be defined.

Line 1096: "Figure S9" was corrected but not fully corrected!

Reviewer #3: I have only a few minor comments that need not necessarily be addressed.

• Metabolites. It is surprising that there are not cascading effects of the allele across all metabolites in this pathway. That is, if the allele impacts the concentration of a metabolite upstream, then one would naively expect downstream metabolite concentrations to also be impacted. Also, wouldn't the K267E substitution be predicted to increase, not decrease TBZ-OH abundances? (I may be wrong about that) Anyway, it is suggested that "additional detoxification" mechanisms are likely to be at play, but could it at all be speculated in the discussion what might be going on here?

• Fitness, trade-offs, and cytochrome P450. For me, cytochromes are implicated in the electron transport chain and thus might be expected to have wide-ranging impacts on metabolism. Little did I realize that these proteins are also often implicated in xenobiotic metabolism specifically. While this is mentioned in the introduction, I think fleshing this out a little bit more in the discussion may be warranted. Mainly because, without this context, the lack of fitness trade-offs connected to this allele remain unexplained. That is, if CYP-35D1 were expected to be pleiotropic, one might expect the lysine allele to be at a disadvantage in the absence of thiabendazole. Along the same lines-- how many CYP proteins are present in the C. elegans genome? If there is a lot of copy number variation, then the opportunity for redundancy and neofunctionalization is high, which would also help to explain the apparent lack of a trade-off here. And an additional minor point, I was surprised that so few Caenorhabditis species were included in the phylogenetic analysis of CYP-35D1 in Figure 7 as there are many genomes now available (maybe there are species out there with the 267 lysine?). Alternatively, I might have missed that this gene only has single-copy orthologs in the reported species; in which case, a larger gene family tree might better contextualize this result.

• Data and code availability. While it is claimed that the data and code have been made available (https://github.com/AndersenLab/2023_cyp35d1_TBZmanuscript), this link leads to a 404 error message. I also did not find any NCBI accession/bioproject numbers associated with this work. I do appreciate the extensive sharing of data in the supplemental materials.

• Figure 1A. This figure might be clearer if, in the legend or the figure itself, it is noted that body length is being used as the proxy for "response" depicted here.

• Line 158. The Andersen et al. 2015 paper should be cited here to clarify that these RILs were previously constructed.

• Lines 473-480. Were the guide RNA sequences reported in these cited papers? Or, I may have missed these somewhere.

Thanks for sharing this work!

PLOS authors have the option to publish the peer review history of their article (what does this mean?). If published, this will include your full peer review and any attached files.

Reviewer #1: No

Reviewer #2: No

Reviewer #3: No

---

## [Editor Report · Decision Letter 1]

29 Dec 2024

Dear Dr. Andersen,

We are pleased to inform you that your manuscript 'Naturally occurring variation in a cytochrome P450 modifies thiabendazole responses independently of beta-tubulin' has been provisionally accepted for publication in PLOS Pathogens.

Best regards,

Adler R. Dillman, Ph.D.

Academic Editor

PLOS Pathogens

James Collins III

Section Editor

PLOS Pathogens

Sumita Bhaduri-McIntosh

Editor-in-Chief

PLOS Pathogens

orcid.org/0000-0003-2946-9497

Michael Malim

Editor-in-Chief

PLOS Pathogens

orcid.org/0000-0002-7699-2064

Thank you for your detailed response to reviewers. It was clear that all reviewers appreciated the importance of this work, but they also offered salient concerns about some details of the first submission. The revised manuscript is a significant improvement.
---

## [Editor Report · Acceptance letter]

7 Jan 2025

Dear Dr. Andersen,

We are delighted to inform you that your manuscript, "Naturally occurring variation in a cytochrome P450 modifies thiabendazole responses independently of beta-tubulin," has been formally accepted for publication in PLOS Pathogens.

Best regards,

Sumita Bhaduri-McIntosh

Editor-in-Chief

PLOS Pathogens

orcid.org/0000-0003-2946-9497

Michael Malim

Editor-in-Chief

PLOS Pathogens

orcid.org/0000-0002-7699-2064